# Sex differences in microglial CX3CR1 signalling determine obesity susceptibility in mice

Mauricio D. Dorfman[1], Jordan E. Krull[1], John D. Douglass[1], Rachael Fasnacht[1], Fernando Lara-Lince[1], Thomas H. Meek[1], Xiaogang Shi[1], Vincent Damian[1], Hong T. Nguyen[1], Miles E. Matsen[1], Gregory J. Morton[1] & Joshua P. Thaler[1]

Female mice are less susceptible to the negative metabolic consequences of high-fat diet feeding than male mice, for reasons that are incompletely understood. Here we identify sex-specific differences in hypothalamic microglial activation via the CX3CL1-CX3CR1 pathway that mediate the resistance of female mice to diet-induced obesity. Female mice fed a high-fat diet maintain CX3CL1-CX3CR1 levels while male mice show reductions in both ligand and receptor expression. Female Cx3cr1 knockout mice develop 'male-like' hypothalamic microglial accumulation and activation, accompanied by a marked increase in their susceptibility to diet-induced obesity. Conversely, increasing brain CX3CL1 levels in male mice through central pharmacological administration or virally mediated hypothalamic overexpression converts them to a 'female-like' metabolic phenotype with reduced microglial activation and body-weight gain. These data implicate sex differences in microglial activation in the modulation of energy homeostasis and identify CX3CR1 signalling as a potential therapeutic target for the treatment of obesity.

[1] UW Diabetes Institute and Department of Medicine, University of Washington, Seattle, Washington 98109, USA. Correspondence and requests for materials should be addressed to J.P.T. (email: jpthaler@uw.edu).

Obesity is a global epidemic that is the leading risk factor for the development of adverse metabolic comorbidities including type 2 diabetes[1]. Previous studies have shown that diet-induced obesity (DIO) is associated with hypothalamic neuronal injury and gliosis in both rodents and humans[2–4]. This response, evident in male rodents from the onset of high-fat diet (HFD) feeding, includes the accumulation of reactive, pro-inflammatory microglia (the resident central nervous system (CNS) macrophages) in the hypothalamus. Unlike male mice, females are generally more resistant to DIO[5–8] and show neither HFD-induced hypothalamic inflammation nor microgliosis[9], suggesting that sex-based differences in microglial responsiveness may contribute to differential DIO susceptibility.

Microglia are the resident immune cells of the brain. Their long cellular projections (dendrites) perform constant surveillance of the CNS parenchyma, serving to detect invading pathogens, circulating biomolecules and host-derived damage signals arising from neuronal stress[10,11]. In response to these cues, microglia become activated, adopting a more amoeboid cellular morphology and producing secreted factors such as cytokines that influence neuronal viability[11]. While this response can serve a protective function, sustained activation of microglia as observed in many neurodegenerative diseases can potentially cause neuronal dysfunction[12]. Similarly, hypothalamic microglia become activated during HFD feeding[3,4], but whether this response promotes or prevents DIO is currently unknown.

To limit the destructive capacity of microglia under basal conditions, neurons secrete a variety of inhibitory factors including the chemokine CX3CL1 (also referred to as fractalkine), a cleavable transmembrane protein that binds the $G_i$-protein coupled receptor CX3CR1 on the surface of microglia[13]. CX3CR1 was originally identified in lymphocytes and is involved in immune regulation in a variety of tissues such as bone, kidney and the cardiovascular system[14–16]. However, CX3CR1 function has been most extensively studied in microglia[17–20], which have >1,000-fold higher level of CX3CR1 expression compared with peripheral myeloid cells and other CNS cell types including neurons and astrocytes[21,22]. Studies of $Cx3cr1^{gfp/gfp}$ knock-in mice, in which the $Cx3cr1$ gene has been replaced by $Gfp$, have revealed that CX3CR1-deficient microglia show excessive cellular activation and overproduction of inflammatory mediators, generally increasing susceptibility to CNS inflammatory diseases[17,23]. However, some studies have identified protective aspects of CX3CR1 deficiency[18], highlighting the complex multifaceted role of microglia in CNS regulation.

In this study, we manipulated microglial activation through CX3CR1 signalling to investigate the causal relationship between microglial reactivity and DIO sensitivity using the opposite responses of male and female mice as a model system[24–26]. Analysis of metabolic parameters and hypothalamic microglial profiles of mice exposed to HFD revealed not only sex differences in DIO susceptibility and microglial activation but also regulation of CX3CL1 and CX3CR1 expression, both of which were reduced in males only. Accordingly, CX3CR1-deficient (knockout (KO)) male mice were phenotypically indistinguishable from controls while female KOs showed 'male-like' microglial and metabolic responses to HFD feeding. Conversely, increasing CX3CL1-CX3CR1 signalling in males in the whole CNS or hypothalamus alone reduced their HFD-induced microgliosis and DIO susceptibility. Together, these findings reveal a contribution of microglial signalling to sex differences in metabolism and support the development of microglial activity modulators for obesity therapy.

## Results

**Females are resistant to DIO and microglial activation.** In rodents, males consuming HFD develop obesity associated with a hypothalamic injury response that includes increased microglial activation and inflammatory signalling[3,4]. In contrast, female mice are more DIO resistant[5,6], but their hypothalamic responses to HFD have been largely uncharacterized. To determine if microglial responses to HFD feeding are sex-specific, we studied cohorts of male and female wild-type (WT) C57BL6/J mice matched for body composition parameters (fat mass, lean mass; Supplementary Fig. 1a,b) and subsequently fed chow or 60% HFD ad libitum for 18 weeks. Consistent with previous reports[27,28], body weights of HFD-fed males significantly diverged from their chow-fed controls as early as 4 weeks after HFD initiation (Fig. 1a). In contrast, WT females were more resistant to DIO with only a small difference ($\sim 3.5$ g) between chow and HFD-fed groups at the end of the dietary intervention (Fig. 1a; female HFD versus chow, $P = 0.09$ by two-way analysis of variance (ANOVA) with post hoc Bonferroni testing). This level of DIO resistance in female mice has been previously reported in the literature (refs 5,6,8) but may have been increased from our use of single housing and strict age matching within cohorts. Cumulative food intake did not differ between diet treatments of either sex (Fig. 1b), suggesting that weight gain results from a reduction of metabolic rate and/or a failure to lower caloric intake to match expenditure. The excess body-weight gain in HFD-fed males was largely a result of increased fat mass (Fig. 1c; per cent fat mass in Supplementary Fig. 3a), since lean mass was minimally affected by diet composition (Fig. 1d; per cent lean mass in Supplementary Fig. 3b). In contrast, females fed the same HFD had small increases in fat mass (Fig. 1c; Supplementary Fig. 3a) that were offset by nearly equivalent reductions in lean mass (Fig. 1d; Supplementary Fig. 3b). Consistent with enhanced DIO susceptibility in males, glucose tolerance after 18 weeks was impaired only in males (Fig. 1e,f), likely as a consequence of increased fat mass.

HFD exposure induces hypothalamic expression of inflammatory cytokines such as IL-1, IL-6 and TNF-α, as well as NF-κB pathway genes ($Nfkbia$ and $Ikbkb$) in male mice[3,4,9,29]. In contrast, female mice have been reported to show no changes in hypothalamic expression of $Il-1$, $Il-6$ and $Tnf-\alpha$ after HFD feeding[9]. To investigate the contribution of microglial cells to HFD-induced hypothalamic inflammation, we extracted microglia from the hypothalamus of a separate cohort of male and female mice exposed to HFD or chow for 3 months, when differences in weight gain are already evident in males. While hypothalamic microglia from HFD-fed males showed increased $Il-1b$ and $Ikkb$ expression compared to chow-fed controls (Fig. 1g), microglia from HFD-fed female mice had no evidence of increased inflammatory gene expression (Fig. 1h). Consistent with these sex differences, levels of CX3CL1 increased in the hypothalamus of females exposed to HFD, while decreasing in males (Fig. 1i). At the receptor level, $Cx3cr1$ messenger RNA (mRNA) expression decreased in hypothalamic microglia of male mice exposed to HFD, but remained unchanged in female mice (Fig. 1j,k). These data suggest that reduced CX3CL1-CX3CR1 signalling in males during HFD feeding could account for sexual dimorphism in HFD-induced microglial activation.

**Female mice lacking CX3CR1 are more susceptible to DIO.** To explain the link between the absence of hypothalamic microglial activation (Fig. 1h) and relative protection from DIO observed in female mice (Fig. 1a), we hypothesized that intact CX3CR1 signalling in females during HFD feeding (Fig. 1i) limits microglial reactivity and reduces obesity susceptibility. By the same logic, male mice would be relatively unaffected by loss of CX3CL1-CX3CR1 signalling since gene expression of both the ligand and receptor are already reduced by HFD exposure. To test

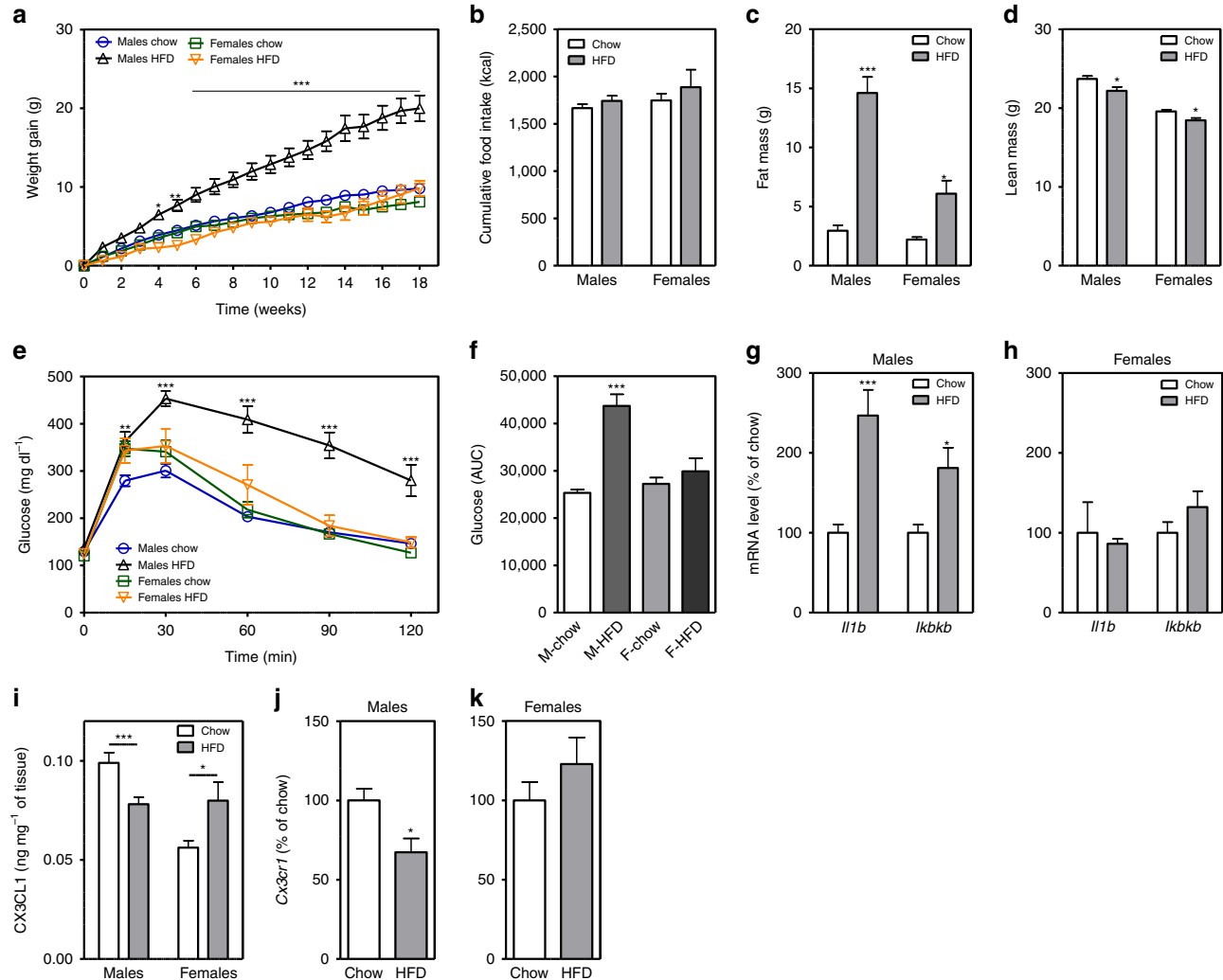

**Figure 1 | Female mice are protected from diet-induced obesity and hypothalamic microglial activation.** (**a**,**b**) Body-weight gain (**a**) and cumulative food intake (**b**) in male and female mice exposed to chow or HFD for 18 weeks. (**c**,**d**) Total fat mass (**c**) and lean mass (**d**) in chow-fed and HFD groups. See also Supplementary Fig. 1. (**e**) Intraperitoneal glucose tolerance test ($2\,g\,kg^{-1}$) in male and female mice after 18 weeks on chow or HFD. (**f**) Glucose AUC analysis of data from **e**. M, males; F, females. Data are presented as mean ± s.e.m. of 12 males and 8 females per group in **a**–**f**. (**g**,**h**) mRNA levels of inflammatory markers in isolated hypothalamic microglia following 3-mo chow or HFD exposure (**g**: males; **h**: females). Data are presented as % change relative to chow-fed controls. Mean ± s.e.m., $n = 8$. *$P < 0.05$ by Student's $t$-test comparing within gender (**i**) CX3CL1 protein expression analysed by ELISA in homogenized hypothalamus dissected from male and female mice maintained on chow or HFD for 18 weeks. Data are presented as mean ± s.e.m., $n = 8$. (**j**,**k**) mRNA level of $Cx3cr1$ in isolated hypothalamic microglia following 3-mo chow or HFD exposure. Data are presented as per cent change relative to chow-fed controls. Mean ± s.e.m., $n = 8$. *$P < 0.05$ by Student's $t$-test comparing within gender. For all panels (except **g**,**h**,**j**,**k**), data are analysed by repeated measures or two-way ANOVA followed by Bonferroni *post hoc* comparisons. *$P < 0.05$, **$P < 0.01$ and ***$P < 0.001$.

these suppositions, we performed metabolic studies in both male and female $Cx3cr1^{gfp/gfp}$ knock-in mice (KO), a model of CX3CL1-CX3CR1 signalling deficiency characterized by hyperactivated microglia[17,30]. Initially, we verified that $Cx3cr1^{gfp/+}$ heterozygotes (HT) and WT mice were metabolically equivalent to allow for the use of HT and KO littermates and facilitate microglial analysis using the green fluorescent protein (GFP) marker. Both on regular chow and during HFD feeding, male and female HTs and WTs gained equivalent weight (Supplementary Fig. 2a). Subsequently, we studied cohorts of male and female HT and KO mice matched for initial body composition parameters (Supplementary Fig. 1c–f) and fed chow or HFD for 18 weeks (Fig. 2). As predicted and consistent with most previous studies[31,32], male KOs were phenotypically indistinguishable from HT littermates

irrespective of diet in terms of weight gain (Fig. 2a; absolute body weight in Supplementary Fig. 2b), food intake (Fig. 2c), fat mass (Fig. 2e; per cent fat mass in Supplementary Fig. 3c), lean mass (Fig. 2g; percent lean mass in Supplementary Fig. 3e) and energy expenditure (HT versus KO adjusted heat production: $0.0248 \pm 0.0016\,kcal\,h^{-1}\,g^{-1}$ versus $0.0243 \pm 0.0018\,kcal\,h^{-1}\,g^{-1}$). In contrast, compared to both chow-fed KO and HFD-fed HT control littermates, female KO mice manifested a more 'male-like' response to HFD consumption with markedly increased body-weight gain (Fig. 2b; Supplementary Fig. 2c) and adiposity (Fig. 2f; Supplementary Fig. 3d) but not lean mass (Fig. 2h; note, slight reduction in HFD-fed HT versus chow-fed HT mice; Supplementary Fig. 3f). In fact, the magnitude of the weight increase was nearly equivalent to that of males (compare Fig. 2a

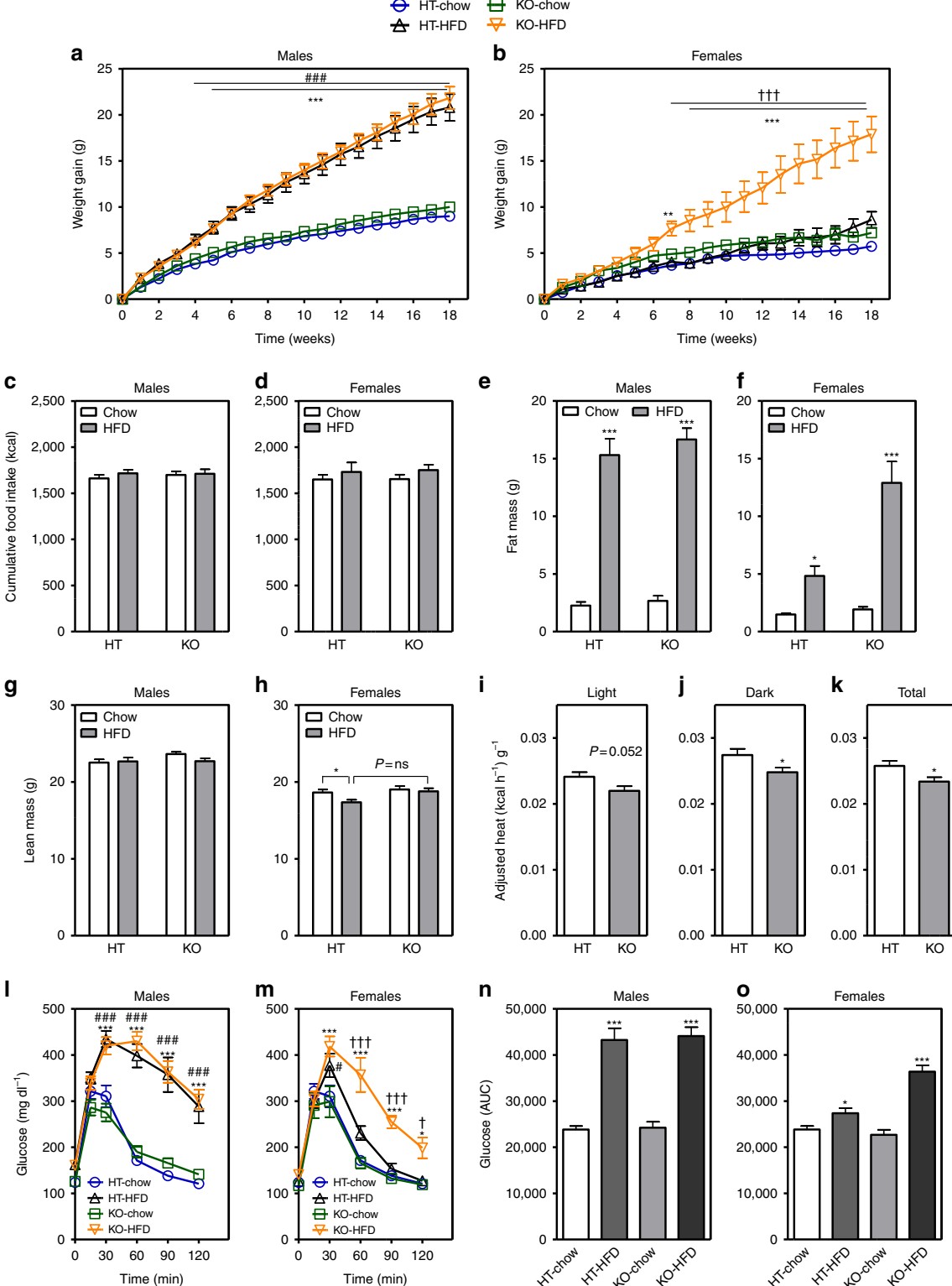

**Figure 2 | *Cx3cr1*-deficient mice exhibit a female-specific obesity phenotype.** (**a**–**d**) Body-weight gain (**a**,**b**) and cumulative food intake (**c**,**d**) measured in *Cx3cr1*-HT and KO mice on chow and HFD for 18 weeks (**a**,**c**: males; **b**,**d**: females). (**e**–**h**) Total fat mass (**e**,**f**) and lean mass (**g**,**h**) measured at study end in HT and KO mice (**e**,**g**: males; **f**,**h**: females). (**i**–**k**) Average adjusted heat production during 12h light cycle (**i**), 12h dark cycle (**j**) and both photoperiods combined (total) (**k**) in HT and KO female mice exposed to HFD for 18 weeks. Heat production was adjusted using normalization to lean mass + 0.2 × fat mass (see 'Methods' section). (**l**,**m**) Intraperitoneal glucose tolerance test (2 g kg$^{-1}$) in HT and KO mice on chow and HFD for 18 weeks (**l**: males; **m**: females). (**n**,**o**) Glucose AUC analysis of data from (**l**,**m**), respectively. Data are presented as mean ± s.e.m. of 12 males and 8 females per group for all panels except (**i**–**k**) where $n = 8$ females per group. For all panels, data are analysed by repeated measures or two-way ANOVA followed by Bonferroni *post hoc* comparisons. *$P < 0.05$, **$P < 0.01$ and ***$P < 0.001$ versus chow groups; #$P < 0.05$, ##$P < 0.01$, ###$P < 0.001$ HT-HFD compared to HT chow. †$P < 0.05$ and †††$P < 0.001$ KO-HFD versus HT-HFD.

males KO HFD with Fig. 2b females KO HFD). The excess weight gain of female KOs consuming the HFD was not associated with changes of food intake (Fig. 2d), but instead, these mice exhibited reduced energy expenditure during both the light and dark cycles (Fig. 2i–k) with no changes in respiratory quotient (Supplementary Fig. 2d) or ambulatory activity (Supplementary Fig. 2e). To rule out the possibility that 'male level' weight gain observed in HFD-fed female KOs involved loss of female gonadal function[5], we verified the presence of normal ovarian structure, oestrous cyclicity and estradiol (E2) production (Supplementary Fig. 4a–d). Finally, male mice develop glucose intolerance during HFD feeding irrespective of genotype (Fig. 2l,n), while only females deficient in CX3CR1 become substantially glucose intolerant (Fig. 2m,o). Thus, though more studies are needed, the effect of CX3CR1 deletion to impair glucose homeostasis likely occurs as a secondary consequence of increased fat mass with HFD feeding. Overall, these results suggest that intact CX3CR1 signalling during HFD feeding confers protection from DIO to female mice.

**HFD-fed *Cx3cr1* KO females display hypothalamic inflammation.**
Male rodents consuming HFD display a hypothalamic injury response that includes accumulation of activated pro-inflammatory microglia[3,4], whereas females are largely resistant to these changes. To quantify microglial number in KO mice, we used immunohistochemical staining for the specific microglial marker, Iba1. In male KOs, CX3CR1 deficiency did not exacerbate further the hypothalamic microgliosis induced by the HFD (Fig. 3a–e). In contrast, HFD-fed female KOs showed expansion of the mediobasal hypothalamus (MBH) microglial population to levels comparable to HFD-fed males, an effect that was not observed in HFD-fed HT females (Fig. 3f–j). This increase in hypothalamic microglial number was verified by counting GFP-positive cells in HT and KO female mice (Supplementary Fig. 5), yielding nearly identical results.

Compared with resting cells, activated microglia adopt a more amoeboid morphology with shorter, thickened dendritic projections[11]. As a result, there is an inverse correlation between microglial process length and the degree of activation. A morphometric analysis of microglia in the MBH of KO and HT females revealed that HFD feeding caused a significant reduction in total dendrite length in KO females but not HT controls (Fig. 3k–m). Consistent with this morphologic indication of cellular activation, hypothalamic microglia from female KOs on HFD also displayed evidence of increased inflammatory markers compared to HT controls (Fig. 3n), again reminiscent of the male response to HFD feeding[3,4] (Fig. 1g). These data collectively support the hypothesis that loss of CX3CR1 predisposes female mice to HFD-induced microglial accumulation and activation in the MBH.

**Oestrogen does not mediate sex differences in *Cx3cr1* KO mice.**
As described above, CX3CL1-CX3CR1 signalling is down-regulated by HFD feeding in males but remains intact in females. As a result, only female KOs are affected by CX3CR1 deficiency with increased HFD-induced microglial activation and weight gain. Therefore, we next addressed the possibility that oestrogen mediates these sex differences in CX3CR1 signalling and DIO susceptibility. First, we performed ovariectomy (OVX) or sham-operation on WT female mice (Supplementary Fig. 6a–d). One week after surgical recovery, mice were fed HFD for 2 weeks, a short exposure chosen to minimize confounding by differences in weight gain (Supplementary Fig. 6a) but allow for the onset of oestrogen deficiency (evident by decreased uterine weight; Supplementary Fig. 6b). The levels of *Cx3cl1* and *Cx3cr1* mRNA

in the hypothalamus did not differ between OVX and sham-operated females, suggesting oestrogen is not a significant physiologic regulator of *Cx3cl1* and *Cx3cr1* gene expression. Similarly, pharmacologic administration of E2 to male mice did not alter *Cx3cl1* and *Cx3cr1* gene expression (Supplementary Fig. 6e,f).

Next, to assess the functional relationship between oestrogen and CX3CR1 signalling, we combined OVX with HFD feeding in a cohort of female KO and HT controls (Supplementary Fig. 6g–j). OVX KO and HT animals became equally obese and hyperphagic after 9 weeks of HFD exposure (Supplementary Fig. 6g,h), indicating that DIO in the context of OVX is unaffected by CX3CR1 deficiency. This result suggested that oestrogen deficiency either promotes weight gain through inactivation of microglial CX3CR1 or uses a parallel pathway that engages common downstream mechanisms such as increased production of microglial inflammatory mediators. To discriminate between these two possibilities, we restored oestrogen levels in the 9 week HFD-fed OVX cohort with continuous E2 infusion via subcutaneously-implanted silastic capsules. Four weeks of E2 treatment caused a marked reduction in body weight and food intake that was equivalent between OVX KO and HT mice (Supplementary Fig. 6i,j), suggesting that oestrogen and CX3CR1 protect against DIO via independent mechanisms. To confirm this finding using a different paradigm, we measured E2-mediated anorexia in HFD-fed male KO and HT mice. A single injection of the long-acting estradiol benzoate (10 µg s.c.) decreased 72 h food intake, but this effect did not differ between genotypes (Supplementary Fig. 6k), consistent with the results from the E2-treated OVX females. Together, these data suggest oestrogen is not the primary mediator of sex-specific protection from DIO through the CX3CR1 signalling system.

**Hypothalamic CX3CL1 reduces body weight during HFD feeding.**
Though CX3CR1 deletion only affected DIO susceptibility in female mice (Fig. 2), this may reflect the already reduced CX3CL1 and CX3CR1 levels in the hypothalamus of HFD-fed male mice (Fig. 1i,j). Thus, it is plausible that central administration of CX3CL1 to male mice might prevent hypothalamic gliosis (as in females) and thereby reduce HFD-induced weight gain. To test this possibility, we used a pharmacologic approach in two settings: a trial to prevent HFD-induced weight gain and another to treat established DIO. Initially, we performed continuous intracerebroventricular (ICV) infusion of the CX3CR1 ligand CX3CL1 (500 ng per day) in male HTs and KOs fed chow diet to test for potential toxicity and off-target effects. As expected, central administration of CX3CL1 did not affect body-weight gain (Supplementary Fig. 7a) and food intake in either HT or KO mice over the 10-day chow diet period (Supplementary Fig. 7b). However, on switch to HFD, HT mice had an immediate reduction of weight gain (Supplementary Fig. 7a,c) and caloric intake compared to KO mice (Supplementary Fig. 7d). In a longer follow-up study, central CX3CL1 limited HFD-associated weight gain (Fig. 4a) and decreased food intake over the first 5 days of the infusion (Fig. 4b) only in HT mice. Immunostained sections from the MBH of HT mice receiving CX3CL1 demonstrated reduced number of microglial cells in the MBH compared to the KOs (Fig. 4c–e), with microglial number approaching the chow-fed norm (compare Fig. 4e with Fig. 3e).

In the second pharmacologic approach, we assessed the efficacy of CX3CL1 as an obesity treatment using an established DIO model of WT C57BL6 male mice fed HFD for 3 months. Weight-matched animals received either daily CX3CL1 (1 µg per day) or vehicle (saline) ICV injections for 28 days. While both groups lost ~1 g during the first 3–4 days of treatment, CX3CL1-treated mice continued to lose weight while vehicle-treated

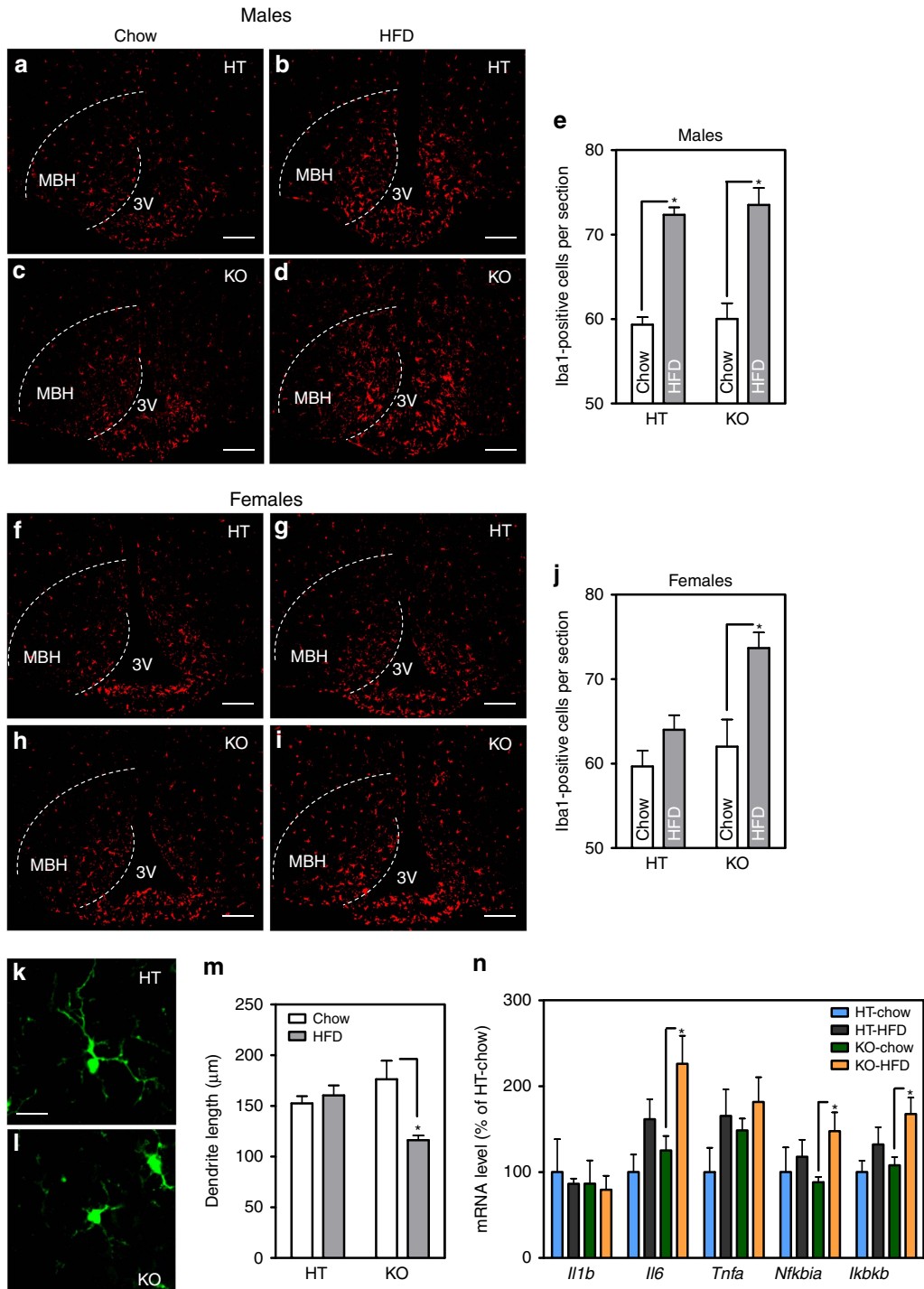

**Figure 3 | *Cx3cr1*-deficient female mice display HFD-induced hypothalamic microglial accumulation and activation.** (**a**–**d**,**f**–**i**) Representative images showing Iba1 immunoreactivity in the MBH of male (**a**–**d**) and female mice (**f**–**i**). 3V = third ventricle. (**e**,**j**) Quantification of Iba1-positive cells in bilateral MBH from six sections per animal. Mean ± s.e.m., $n = 4$ per group. Scale bar, 50 μm. (**k**,**l**) Representative images showing microglial morphology in the MBH of HT and KO females. Scale bar, 10 μm. (**m**) Quantification of total dendritic length in MBH from 10 individual microglial cells per animal. Mean ± s.e.m., $n = 4$ per group. (**n**) mRNA levels of inflammatory markers in isolated hypothalamic microglia following 12-week HFD exposure in HT and KO female mice. Data are presented as per cent change relative to HT-chow controls. Mean ± s.e.m., $n = 5$–7 per group. For all panels, data are analysed by two-way ANOVA followed by Bonferroni *post hoc* comparisons. *$P < 0.05$.

animals recovered and gained weight from day 15 onward (Fig. 4f; asterisk denotes statistically significant between-group comparisons). Body composition analysis performed 2 days before and during the ICV injection period (day 22) revealed that the weight loss resulted from decreased fat mass (Fig. 4g) since

lean mass was unchanged between groups (CX3CL1 versus Veh: 19.81 ± 0.40 g versus 20.52 ± 0.36 g; $P = 0.17$ by Student's *t*-test). Consistent with the prevention studies (Fig. 4a,b and Supplementary Fig. 7), CX3CL1 treatment lowered food intake (Fig. 4h), likely accounting for the body-weight reduction.

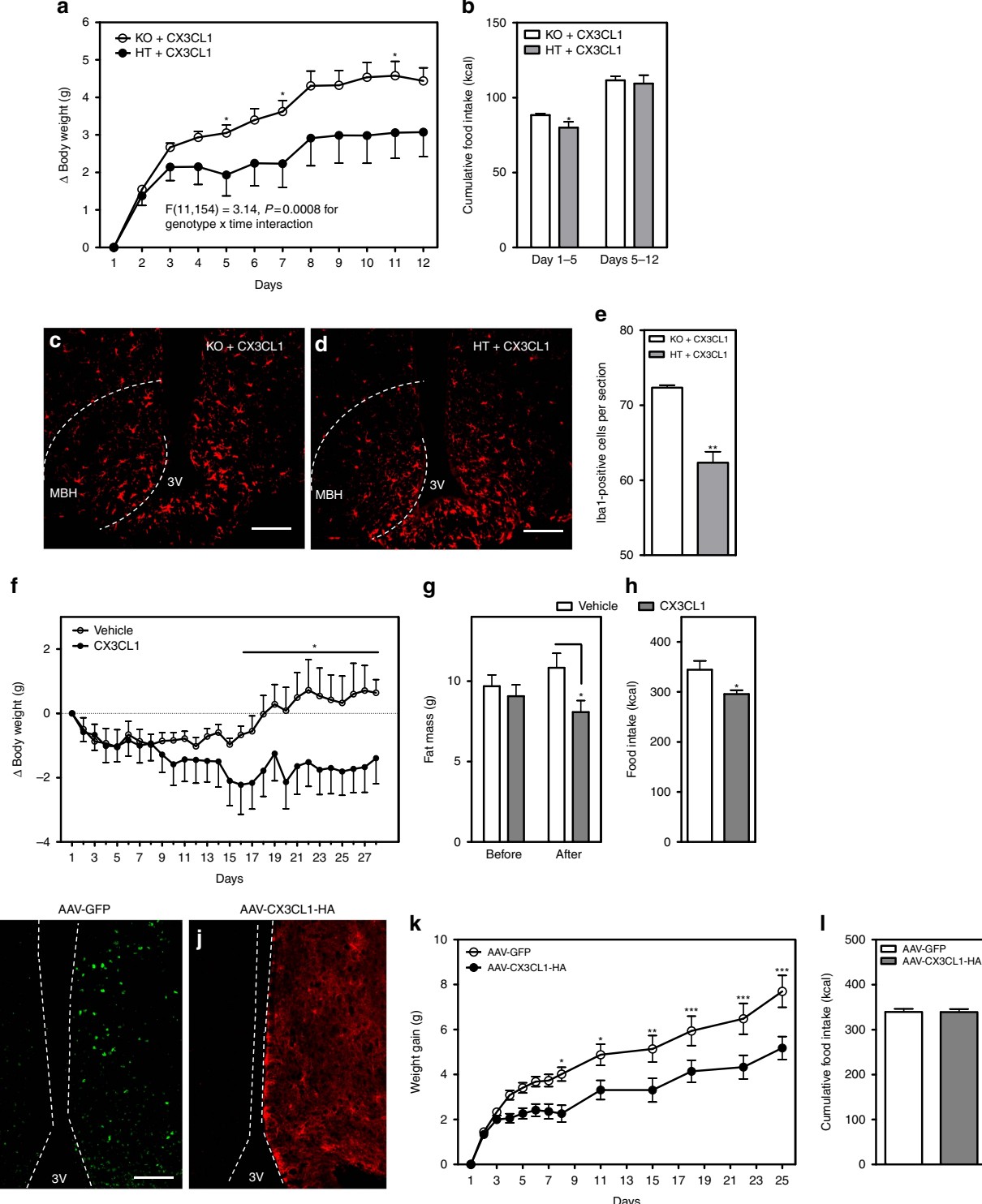

**Figure 4 | Central administration and hypothalamic viral overexpression of CX3CL1 limits HFD-induced weight gain in males.** (**a,b**) Body-weight gain (**a**) and cumulative food intake (**b**) in HFD-fed *Cx3cr1* KO and HT male mice infused ICV with CX3CL1 (500 ng per day) over 12 days. $F_{(11,154)} = 3.14$, $P = 0.0008$ for genotype × time interaction using repeated measures ANOVA. Mean ± s.e.m., $n = 6$–10 per group. (**c,d**) Representative images showing Iba1 immunoreactivity in the MBH of KO (**c**) and HT male mice (**d**) centrally treated with CX3CL1 for 12 days. Scale bar, 50 μm. (**e**) Quantification of Iba1-positive cells in bilateral MBH from six sections per animal. Mean ± s.e.m., $n = 4$ per group. (**f**) Body-weight change in 3-month DIO WT males injected ICV daily with CX3CL1 (1 μg per day) or vehicle (saline) over 28 days. (**g**) Total fat mass measured 2 days before and after 3 weeks of ICV injections (day 22). (**h**) Cumulative food intake measured over 28 days of ICV vehicle or CX3CL1 treatment. Mean ± s.e.m. of eight animals per group in (**f–h**). (**i,j**) Representative images of the MBH from mice injected unilaterally for validation with AAV-GFP (**i**) or AAV-CX3CR1-HA (**j**). Scale bar, 50 μm. (**k,l**) Body-weight gain (**k**) and cumulative food intake (**l**) in HFD-fed mice injected bilaterally in the MBH with AAV-GFP or AAV-CX3CL1-HA. Mean ± s.e.m. of nine animals per group. For **a,f,k** data are analysed by repeated measures ANOVA followed by Bonferroni *post hoc* comparisons. \*$P < 0.05$, \*\*$P < 0.01$. \*\*\*$P < 0.001$.

Finally, we assessed whether overexpressing CX3CL1 specifically in the MBH could reduce weight gain and caloric intake. Groups of weight-matched male mice received intra-MBH injections of adeno-associated virus (AAV) expressing either full-length CX3CL1 with an hemagglutinin (HA) tag[33] or GFP. After 2 weeks to allow for viral expression, injected mice were switch to HFD for a period of 28 days (Fig. 4i–k). Remarkably, CX3CL1 overexpression in the MBH limited weight gain on HFD relative to AAV-GFP-injected controls (Fig. 4i–k). Unlike the CX3CL1 ICV infusion studies, however, caloric intake was unchanged between groups (Fig. 4l), suggesting that CX3CL1 action in the hypothalamus primarily regulates energy expenditure. Overall, these data demonstrate that restoring hypothalamic CX3CL1-CX3CR1 signalling in male mice promotes a 'female-like' response to HFD feeding characterized by both reduced microglial activation and DIO susceptibility.

## Discussion

In this study, we have taken advantage of the differing gliosis and DIO susceptibility of male and female C57BL6 mice as a model system to determine the reciprocal effects of microglial activation (CX3CR1 KO) and suppression (CX3CL1 treatment) on energy homeostasis. We found that hypothalamic microglial activation and inflammatory signalling not only correlate with but confer sensitivity to HFD-associated weight gain. Accordingly, HFD induction of increased CX3CR1 signalling in females affords them protection from DIO through reduced microglial activation while males lack this adaptive mechanism due to downregulation of CX3CL1 and *Cx3cr1* during HFD feeding. With CNS administration or MBH viral overexpression of CX3CL1 to reverse this deficit, DIO-sensitive males adopt a more resistant 'female-like' phenotype with reduced microgliosis and weight gain. Thus, microglial CX3CR1 signalling represents a molecular switch that toggles the degree of obesity susceptibility in a sex-specific manner.

Most previous studies examining the role of hypothalamic gliosis in DIO susceptibility focused exclusively on male mice[2–4]. In this study, we have included both males and females, and found that the marked variation in DIO susceptibility between the sexes correlates with their differential sensitivity to HFD-induced microglial activation. Though disparities in CX3CL1 and CX3CR1 levels may contribute to these sex differences, the results also suggest the possibility of microglial sexual dimorphism. Indeed, microglia not only participate in establishing the classically dimorphic preoptic area (reviewed in ref. 34), but also show sex-specific responses to inflammatory stimuli *in vitro*[35]. Likewise, excess hypoxia/ischaemia-induced microglial activation in CX3CR1-deficient females but not males results in hippocampal injury and learning deficits[36]. Finally, several reports have highlighted intrinsic sex differences in microglial activity or gene regulation[25,26]. Though oestrogen is a potent anti-obesity hormone[37,38] that can suppress microglial activation and inflammatory signalling[39–41], our data argue against a primary role for oestrogen in mediating CX3CR1-associated sex differences. Nevertheless, other unidentified sex-specific microglial factors may account for the contrasting alterations to energy homeostasis parameters observed with CX3CR1 deficiency in females (energy expenditure) versus CX3CL1 administration in males (food intake). Alternatively, regional variation in microglial function[42] may impart distinct metabolic consequences to altering microglial activation in different brain areas, potentially explaining the sex differences as well as the disparate effects of ICV infusion and hypothalamic overexpression of CX3CL1 on food intake. Indeed, important aims for future study include determining the mechanism

underlying the diet-specific regulation of CX3CL1 and CX3CR1 in males, distinguishing microglial populations that modulate food intake from those that affect energy expenditure, and moving beyond oestrogen to identify epigenetic, immunologic or chromosomal mediators of the sex differences in microglial activation and DIO susceptibility.

In terms of metabolic outcomes, previous reports using a CX3CR1-deficient model have documented reduced atherosclerosis (on an *ApoE*-null background)[43,44] and disrupted beta cell function[31] but have generally not found differences in energy balance or insulin sensitivity[32,45,46] though none of these investigations included female mice. In contrast, a recent study using short interfering RNA-mediated knockdown of hypothalamic CX3CL1 in male HFD-fed Swiss-Webster mice found reduced inflammation and weight gain over the subsequent 6 weeks, similar to our finding with increasing CX3CL1 (ref. 47). These differences may relate to strain (Swiss-Webster is outbred and extremely DIO-sensitive) and potential off-target toxicity of short interfering RNA. The lack of a CX3CL1 treatment effect in the CX3CR1 KOs indicates that CX3CL1-induced reduction in microgliosis and DIO susceptibility reflects on-target silencing of microglia, a finding consistent with the known downstream action of CX3CR1 to inhibit cellular activation through a Gi-coupled mechanism[48]. In addition, CNS administration or viral overexpression of the soluble isoform of CX3CL1 diminishes brain damage in ischaemia and Parkinson's disease models[49–51], consistent with the reduction of HFD-induced gliosis and weight gain observed in our experimental models. Finally, several *Cx3cr1* polymorphisms associated with cardiovascular disease risk in humans have recently been linked with obesity and waist circumference particularly in females[52]. Together, these findings support a role for CX3CR1 signalling in reducing microglial activation and DIO susceptibility.

Until recently, the field of energy homeostasis has been dominated by research into neuronal mechanisms that govern the regulation of food intake and energy expenditure. While neurons are the ultimate output cells of the system, modulation of energy balance involves the interaction of multiple cell types in the brain. For example, previous reports have demonstrated that deficiency of astrocyte leptin receptor signalling enhances refeeding and ghrelin-mediated hyperphagia[53], hypothalamic tanycytes gate leptin access to the MBH[54], and microglia can alter neuronal firing patterns and affect short-term food intake[3,55]. Our study adds to this literature by providing the first direct evidence that (1) microglia can modulate energy homeostasis over the long term; (2) sex differences in microglial activation contribute to obesity susceptibility; and (3) CX3CR1 represents a potential nonneuronal CNS target for obesity therapeutics.

## Methods

**Animals.** The *Cx3cr1gfp* knock-in mouse strain and WT mice on a C57BL6/J background were purchased from Jackson Laboratory (Stock #005582). Both colonies were expanded in our facility with *Cx3cr1gfp*/ + intercrosses used to provide WT, HT and homozygous (KO) littermates. All animals were housed with *ad libitum* access to water and food in a temperature-controlled room with a 12:12 h light:dark cycle under specific pathogen-free conditions. All procedures were performed in accordance with NIH Guidelines for Care and Use of Animals and were approved by the Institutional Animal Care and Use Committee at the University of Washington.

**Genotyping.** *Cx3cr1gfp* (KO) animals were genotyped by PCR using ear genomic DNA and the following set of 3 primers. Primer 1 (forward for WT allele) 5′-CCC AGA CAC TCG TTG CCT T-3′, primer 2 (common reverse for WT and GFP alleles): 5′- GTC TTC ACG TTC GGT CTG GT-3′, and primer 3 (forward for GFP allele) 5′-CTC CCC CTG AAC CTG AAA C-3′.

**Metabolic phenotyping.** An initial pilot study involved group-housed male and female HT and WT mice (*n* = 5 per group) monitored weekly on chow

until 8 weeks old when they were switched to 60% HFD (D12492; Research Diets) for 9 weeks. The primary comprehensive metabolic study involved single-housed male and female KO and HT littermates ($n = 12$ per group and 8 per group, respectively). At 7 weeks old, half of the groups were switched to 60% HFD while the other half remained on normal rodent chow. Body weight and food intake were measured once weekly for an additional 18 weeks. At age 5 weeks (before diet switch) and 24 weeks (study end), body composition analysis was performed using quantitative magnetic resonance spectroscopy (EchoMRI 3-in-1; Echo MRI)[56]. Metabolic phenotypes of HT and KO mice were confirmed with two subsequent cohorts: group-housed males and females ($n = 10$ per group); and single-housed females ($n = 8$ per group).

**Indirect calorimetry.** Animals were acclimated to metabolic cages before measurement of energy expenditure using a computer-controlled indirect calorimetry system (Promethion, Sable Systems). $O_2$ consumption and $CO_2$ production were measured for each animal for 1 min at 10-min intervals as previously described[57–59]. Respiratory quotient was calculated as the ratio of $CO_2$ production to $O_2$ consumption. Energy expenditure was initially calculated using the Weir equation[60]. The conditions for use of ANCOVA analysis[61] were violated by the nonlinear relationships between lean mass and heat production ($r^2 = 0.15$, $P = 0.40$ for HT) and total body mass and heat production ($r^2 = 0.23$, $P = 0.28$ for HT). Linearization of the data were attempted with logarithmic and Box-Cox transformations[62] but did not improve the goodness of fit. Therefore, energy expenditure data were normalized to 'metabolic equivalent weight' (lean mass + C x fat mass) where $C = 0.2$ (based on the findings of Speakman[63] as extended by Even and Nadkarni[64]) since fat mass accounts for about 20% of the energy expenditure of lean mass on a per gram basis. Indeed, this is likely a conservative estimate given the finding of $C = 0.5$ in a large mouse cohort analysed by Kaiyala et al.[61,65]. Ambulatory activity was determined continuously with consecutive adjacent infrared beam breaks in the x-, y- and z-axes scored as an activity count that was recorded every 10 min as previously described[57–59]. Data acquisition and instrument control were coordinated by MetaScreen v.2.0.0.9, and raw data was processed using ExpeData v.1.6.4 (Sable Systems) with an analysis script documenting all aspects of data transformation. Light and dark cycle energy expenditure values are reported based on averaging 72 data points per 12 h cycle on 3 consecutive days, and these in turn were averaged to obtain total 24 h energy expenditure.

**Glucose tolerance test.** Intraperitoneal glucose tolerance tests (30% D-glucose; 2 g kg$^{-1}$ i.p.) were conducted at the end of the 18-week diet period in 5-h fasted mice. Blood glucose levels were measured over 120 min using a hand-held glucometer (OneTouch Ultra) to test tail capillary blood.

**Bilateral ovariectomy and estradiol treatment.** 8-week-old WT female mice were subjected to bilateral OVX or sham surgery ($n = 6$ per group) under isoflurane anaesthesia. After 7 days of recovery, daily measurement of food intake and body weight was recorded for 2 weeks on HFD followed by killing to obtain hypothalamic tissue. A second cohort of 8-week-old mice (HT and KO) subjected to sham or bilateral OVX surgery ($n = 6$ per group) were fed with HFD for 9 weeks before a silastic capsule with 17β-estradiol (E2; 36 µg ml$^{-1}$) was implanted subcutaneously as described by Strom J.O., et al.[66]. The concentration of E2 used was selected to maintain a physiological concentration of circulating oestrogen for 1 month[66]. Weekly measurement of body weight and food intake was recorded before and after E2 treatment.

20-week-old male WT mice ($n = 6$ per group) received a single injection of E2 (1 or 10 µg i.p.) or vehicle (corn oil), and 2 h later the hypothalamus was dissected and frozen at $-80\,°C$.

12-week-old male HT and KO mice ($n = 4–6$ per group) received a s.c. dose of 17β-estradiol-3 benzoate (10 µg) or vehicle (corn oil) at the onset of HFD feeding. Food intake was monitored for the next 72 h.

**CX3CL1 continuous ICV infusion.** Two cohorts of male HT and KOs ($n = 6–10$ per group) underwent lateral ventricle cannulation (Alzet; DURECT Corp) with subsequent implantation of a subcutaneous osmotic minipump connected to the cannula resulting in 14-day continuous ICV infusion of recombinant mouse CX3CL1 (BioLegend; 500 ng per day) or saline vehicle. In the first cohort (age = 8 weeks old), animals were switched to HFD on day 11 while in the second (age = 16 weeks old), HFD was provided from day 1. Daily measurement of body weight and food intake was recorded in both cohorts. At the end of the study, mice were perfused for immunohistochemical outcomes, and the residual volume in each minipump was measured to verify complete infusion.

**CX3CL1 ICV injections in DIO mice.** 1 week after lateral ventricle cannulation at age 8 weeks, WT male C57BL6/J mice received HFD for 3 months with weekly food intake and body-weight measurements. Weight-matched groups were established ($n = 8$ per group), and recombinant mouse CX3CL1 (1 µg in 2 µl) or saline vehicle was injected ICV once daily for 28 days with continued HFD feeding.

**CX3CL1 hypothalamic overexpression.** A vector containing the hybrid CMV-chicken β-actin promoter driving expression of CX3CL1 with an HA-tag appended to the C-terminus[33] (kindly provided by Dr Kevin Nash, University of South Florida) was packaged into a AAV9 vector by the University of Washington Diabetes Research Center Viral Vector and Transgenic Mouse Core. The AAV9-GFP control virus was kindly provided by Dr Michael Schwartz, University of Washington.

Two groups of 8- to 10-week-old male WT C57BL6/J mice were injected with AAV9-CX3CL1-HA or AAV9-GFP (viral titre: $1 \times 10^{12}$ viral particles per ml; $n = 10$ per group). Briefly, mice from each AAV group received two consecutive stereotaxic injections (0.25 µl) bilaterally into the arcuate and ventromedial hypothalamus (anterior–posterior, $-1.4$ mm from bregma; lateral, $\pm 0.5$ mm; dorsal-ventral, $-5.3$ and $-5.7$ mm) using a Hamilton syringe (80030) with a 33-gauge needle at a rate of 50 nl min$^{-1}$ (Micro4 controller) followed by a 7 min waiting period before needle removal. Mice were given 2 weeks to recover and acclimated to handling for 1 week before switching to a HFD. Body weight and food intake were recorded daily for an additional 4 weeks after diet switch. At the end of the study, the location of viral infection/protein expression was verified in all mice by IHC. Two animals in the AAV9-GFP group with staining located outside the MBH were excluded from further analysis.

**Isolation of microglial cells.** Microglial cell extraction was performed as previously described using methods that have been widely validated for obtaining a resting microglial profile[67,68]. Briefly, cohorts of female ($n = 8$ per group) and male mice ($n = 16$ per group) were placed on chow or HFD for 3 months. After perfusion with PBS, whole hypothalamus was dissected and collected in 2 ml of HBSS ($Ca^{2+}$ and $Mg^{2+}$ free) containing 0.25% trypsin (Gibco). Hypothalamic tissues were dispersed into a single-cell suspension using gentle trypsin digestion at $37\,°C$ followed by serum inactivation. After filtration (70 µm), cell suspensions were transferred to a discontinuous Percoll gradient (30, 37 and 70%), and presumptive microglia were collected from the 70–37% interphase after centrifugation. Post-collection, microglial cells were sorted for CD11b or GFP positivity using a FACSAria II Cell sorter (Becton-Dickinson). Microglial cells were collected directly into lysis buffer (RNeasy Micro kit, Qiagen) and stored at $-80\,°C$ until processing.

**Real-time PCR.** Total RNA was extracted using RNeasy micro kit according to manufacturers' instructions (Qiagen) and reverse-transcribed with AMV reverse transcriptase (Promega, WI, USA). For males, hypothalamic microglia RNAs from sets of two mice each were combined to increase the yield per sample (final group $n = 8$). Levels of mRNA for *Il1b*, *Il6*, *Tnfa*, *Nfkbia*, *Ikbkb*, *Cx3cl1*, *Cx3cr1* and *18S* RNA (internal control) were measured by semiquantitative real-time PCR on an ABI Prism 7900 HT (Applied Biosystems). The primer sequences were designed using Primer Express (version 2.0.0; Applied Biosystems) as follows: Il1b fwd: 5′-TAC AAG GAG AGA CAA GCA ACG ACA-3′, rev: 5′-GAT CCA CAC TCT CCA GCT GCA-3′; Il6 fwd: 5′-GTG GCT AAG GAC CAA GAC CA-3′, rev: 5′-GGT TTG CCG AGT AGA CCT CA-3′; Tnfa fwd: 5′-CAT CTT CTC AAA ACT CGA GTG ACA A-3′, rev: 5′-TGG GAG TAG ATA AGG TAC AGC CC-3′; Nfkbia fwd: 5′-TGC CTG GCC AGT GTA GCA GTC TT-3′, rev: 5′-CAA AGT CAC CAA GTG CTC CAC GAT-3′; Ikbkb fwd: 5′-TCA GTG CAT CTC AGA CAG CA-3′, rev: 5′-TAC AGC TGA CAC TTT CCG GT-3′; Cx3cl1 fwd: 5′-CGC GTT CTT CCA ATT TGT GTA-3′, rev: 5′-CTG TGT CGT CTC CAG GAC AA-3′; Cx3cr1 fwd: 5′-CAG CAT CGA CCG GTA CCT T-3′ and rev: 5′-GCT GCA CTG TCC GGT TGT T-3′.

**CX3CL1 ELISA.** CX3CL1 protein concentration (normalized to mg of tissue) in hypothalamic extracts was determined using a mouse CX3CL1 enzyme-linked immunosorbent assay (ELISA) kit based on manufacturer's instructions (R&D Systems).

**Immunohistochemical staining.** Hypothalamic sections were washed with PBS, blocked with 5% normal donkey serum (Jackson Immunoresearch Laboratories), incubated overnight with rabbit anti-Iba1 (1:1,000; Wako Pure Chemicals 019–19741), rabbit anti-HA (1:1,000; Cell Signaling C29F4), chicken anti-GFP (1:5,000; Abcam 13970) and 1 h with appropriate fluorescent secondary antibodies (Alexa Fluor 594-labelled anti-rabbit secondary antibody (1:500; Invitrogen); Alexa Fluor 488-conjugated donkey anti-chicken (1:500; Jackson Immuno Research). Images were captured on an Eclipse E600 upright microscope equipped with a colour digital camera (Nikon).

**Microglial number and dendritic length quantification.** Iba-1 and GFP immunostaining were utilized for microglial quantification. Immunopositive cells were identified in the MBH with localization to arcuate or ventromedial hypothalamic nucleus established using a DAPI counterstain. Subsequently, microglia were counted manually in a blinded fashion on 6 anatomically matched coronal sections per animal using ImageJ (NIH). Replicate values from each animal were individually averaged before determining group means ($n = 4–5$ per group).

For dendritic length measurement, imaging was performed on a Nikon A1R confocal laser scanning microscope using a Plan Apochromat VC 60x WI DIC N2 objective. Z stacks were generated with 0.79 µm steps in z direction, 512 × 512 pixel resolution and analysed using ImageJ software. Ten individual microglial cells were quantified per analysed mouse.

**Oestrous cycle.** Oestrous cyclicity was assessed by light microscopic analysis of daily vaginal lavages to determine the relative proportion of leucocytes, epithelial cells and cornified cells, which characteristically change during the various stages of the oestrous cycle[69].

**Serum estradiol.** Estradiol (E2) levels were determined on serum obtained at killing during diestrus using enzyme immunoassay (ELISA) following the manufacturer's instructions (ES180S-100; Calbiotech). The minimal detectable value of E2 was $< 3$ pg ml$^{-1}$ serum.

**Ovarian morphology.** Ovaries from HT and KO mice were fixed in Kahle's fixative, embedded in paraffin, sectioned at 8 µm, and stained with haematoxylin and eosin. Ovarian sections from the middle of the ovary were analysed by observing corpora lutea and follicles in different developmental stages according to criteria proposed by Pedersen and Peters[70].

**Statistical analyses.** All results are presented as mean ± s.e.m. Statistical significance was determined by unpaired two-tailed Student's $t$-test for two group comparisons. For all diet studies, two-way repeated measures ANOVA with Bonferroni *post hoc* tests were used. Cumulative measures, cell counts and gene expression data were analysed by standard two-way ANOVA with Bonferroni *post hoc* tests. GTT data were also subjected to area under the curve (AUC) analysis. All statistical analyses were conducted with Prism (GraphPad) or Statistica (StatSoft) software. Probability $P$ values of $< 0.05$ were considered significant.

**Data availability.** The data that support the findings of this study are available from the corresponding author upon request.

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

## Acknowledgements

The authors acknowledge the technical assistance provided by J.D. Fisher, Alex Cubelo and Loan Nguyen at the University of Washington, and thank members of the Schwartz laboratory for many fruitful discussions. Dale Hailey at the Mike and Lynn Garvey Cell Imaging Lab (Institute for Stem Cell and Regenerative Medicine, University of Washington) provided assistance with sample imaging and analysis. This work was supported by a mentor-based postdoctoral fellowship from the American Diabetes Association (ADA), a Pilot & Feasibility Award provided by the Nutrition Obesity Research Center (NORC; P30 DK035816) and by the Scientist Development Grant from the American Heart Association to M.D.D., by NIH grant DK089056 to G.J.M., and by an ADA Pathway to Stop Diabetes Grant 1–14-ACE-51 and NIDDK Career Development Award K08 DK088872 to J.P.T. In addition, services and support were provided by the NORC (DK035816) and Diabetes Research Center (DK017047) at the University of Washington.

## Author contributions

M.D.D., J.E.K., J.D.D., R.F., F.L.-L., X.S., V.D., H.T.N. and M.E.M. performed the experiments and collected the data. T.H.M. and G.J.M. assisted with experimental design and analysis. M.D.D. and J.P.T. designed the study, analysed data and wrote the paper.

## Additional information

**Competing financial interests:** The authors declare no competing financial interests related to this manuscript.

