## [Peer Review File · Nature Communications]

Reviewer #1 (Remarks to the Author)

"Gender differences in microglial CX3CR1 signaling determine obesity susceptibility" by Dorfman et al. emphasized the CX3CL1/CX3CR1 as key modulator in DIO associated hypothalamic microgliosis and conclude that higher CX3CL1 input is the main reason that female mice are resistant to DIO. The study was performed carefully and conclusions were drawn properly. This is a timely study on the role played by the non-neuronal cells in regulation of energy metabolism.

My major concern about this study is this study is descriptive without clear mechanism. In fact, authors have proposed several potential mechanistic experiments, that can be performed without obvious obstacles, for example "whether female DIO resistance stems in part from the interaction of CX3CR1 signaling with estrogen, a potent anti-obesity hormone, that also suppresses microglial activation and inflammatory signaling", according to my understanding, the CX3CL1/CX3CR1 interaction with estrogen can be easily checked by ex vivo experiments.

In addition, I have some minor issues:

1. Since most of the experiments were aimed to compare gender or genotype susceptibilities to diet (induced obesity) (interaction), two-way ANOVA but t-test should be performed for all data analysis involving two factors, especially for those in figure.1-3.
2. In Fig.1i and j, basal CX3CL1 levels in Chow diet should not be normalized to 1 in female chow, this failed to show the difference of basal level between male and females. Whether females already had higher CX3CL1 secretion tone under physiological conditions which can prevent the DIO development should be presented. Also, same for the CX3CR1 levels. CX3CR1 protein levels have not been measured in either gender or diet. qPCR data failed to show the CX3CR1 basal mRNA level comparisons between male and female mice. Whether higher tone of CX3CR1 signaling and less iba1 reactivity (in Fig.3 a-j) in female mice are mainly due to higher ligand input, or higher receptor expression, or both should be elaborated.
3. In Fig.2 a and b, the absolute BW from wk0 should be presented, at least in suppl data.
4. The animal model used in this study is a global KO model. Though Fig.4 further tested brain specific effect by ICV infusion, ICV infusion from lateral ventricle could cause whole brain effect. In Fig.1, 3 and 4, at the molecular level, author only showed changes of iba1 immunoreactivity, CX3CL1/CX3CR1 and other gene mRNA levels in medial basal hypothalamus. Whether this global or whole brain phenotype had an exclusive link with medial basal hypothalamus is still questionable. This needs more supporting evidences or references included in the discussion. Besides, whether CX3CL1/CX3CR1 level differences between males and females only take place in MBH is another hole to be filled in Fig.1, or in suppl data. High magnification of iba1-ir images should be presented, to show whether there are differences on iba1-ir morphology, for example the number of processes and coverage of individual microglia.
5. Do neurons/hypothalamic neurons or astrocytes or any other cell populations from male and females have different CX3CL1 secretion levels under Chow and HFD? Does estrogen receptors expressed on the microglia?
6. Whether restore CX3CR1 expression in microglia from HFD fed male brain could rescue the microgliosis?
6. Fig.4 a-e showed preventative effect of central ICV infusion of CX3CL1 to DIO. Here vehicle groups are missing, at least a "HET+VEH" group should be included to show the CX3CL1 effect on WT.
7. All the glucose tolerance data included in the article was explained simply by fat mass difference. These data haven't been interpreted carefully. Whether glucose intolerance was due to fat mass difference needs to be tested in body weight matched studies, brain dominated effects on glucose uptake could not be excluded either.

Reviewer #2 (Remarks to the Author)

This manuscript focuses on the observation that female mice seem protected from the obesity inducing effects of a high fat diet. The hypothesis that activation of microglia in male hypothalamus but not female is the basis for the sex difference in response to a high fat diet is tested and largely confirmed. The notion that brain inflammation is a by product of obesity has been firmly established but there has been relatively little attention to sex differences in the impact of diet and how neuroinflammation might contribute to that difference. The use of a transgenic mouse lacking the fractalkine receptor and administration of the ligand to males creates a well balanced design that confirms differential roles for microglia in diet induced obesity in males and females. Given the human sex differences in obesity this work makes an important contribution to the literature.

There are some conclusions in the manuscript that are not well supported and some concerns about presentation.

Major Concerns.

- 1) Throughout the manuscript and even in the title the authors use the term gender when referring to the mice. Gender is an exclusively human construct that consists of both self and societies perception of ones sex. Animals do not have gender, only sex. Please replace the term throughout.
- 2) The weakest data in the manuscript are on the accumulation of microglia in the hypothalamus of high fat diet fed mice. The use of Iba1 fluorescence staining for counting is not ideal as the sensitivity is low, the signal fades with time and there is no counter stain by which precise anatomical locations can be determined. This would be much better done with DAB immunohistochemistry and proper stereology.
- 3) It would be even better if proliferation of microglia versus the vague term "accumulation" could be discerned by injecting animals with BrdU and colocalizing with Iba1. As is the term "microgliosis" cannot be used since this has not been actually shown.
- 4) The diagram showing the outline of the MBH suggests the focus is on the arcuate, but the MBH includes the VMN as well. A much more precise anatomical characterization of the microglia would enhance the manuscript.
- 5) What is the rationale for standardizing mRNA levels to control in Figure 4K? PCR is a highly quantitative technique and so no % change transformation should be required.
- 6) Doesn't the data in Figure 4 suggest that the effect of the fractalkine receptor activation is to modulate food intake? This seems contrary to the notion that it is metabolism and glucose regulation that is the primary driver of weight gain. It would seem the central hypothesis that microglia activation regulates high fat diet induced obesity would have to be modified to microglia activation regulates appetite for high fat food. There is no effect on food intake in Figure 2 using the KO mouse, so this aspect of the manuscript is very confusing, are the microglia regulating metabolism or food intake?
- 7) The data in Figure 4i are taken to show fat males can loose weight if injected with fractalkine ligand but the line with the asterics is over the animals that received vehicle and gained weight relative to the start. It is not clear what the statistical comparison is here and if it is between or within groups.
- 8) What is missing from the manuscript is any indication of the origin of the observed sex difference. The authors state there is no difference in female gonadal function in the KO mice, but what happens if the gonads are removed in both males and females to remove gonadal steroids? Would the sex difference go away? This simple step would provide major insight as to whether the observed effects are the result of differences in adult hormonal milieu or are the result of either developmental effects of hormones or chromosome complement.

Minor Concerns

- 1) In the Introduction the authors refer to microglia being maintained in a quiescent state at

baseline but this term is no longer used, instead these microglia are surveying as they are actively interacting with nearby neurons all the time and appear to never rest.

2) It is surprising that the act of isolating microglia from the brain did not in and of itself activate them. Others have suggested that this approach cannot be used for assessing short-term signaling in microglia. Can the authors provide any insight as to why they have been successful where others have apparently failed?

3) Pg 4 - last sentence, the phrase "overall lower microglial silencing in males.." is both awkward and inaccurate, there is no evidence microglia are silent in females, they are just not reactive. The next paragraph again uses the term "microglia silencing", this terminology should be removed throughout.

Reviewer #3 (Remarks to the Author)

In this study, Dorfman and colleagues investigated the gender susceptibility to obesity development in the context of microglial CX3CR1 signaling. They show that, in contrast to males, female C57Bl/6 mice are resistant to develop diet-induced obesity. This is associated with reduced hypothalamic microglial activation and increased content of CX3CL1. These results encouraged the authors to assess in more detail the gender effects of CX3CR1 signaling using a model deficient (KO) for this receptor. Interestingly, female KO mice fed with high-fat diet developed obesity to a similar extent than male mice. This "male-like" phenotype observed in KO females, was associated with increased adiposity in the face of normal food intake and reduced energy expenditure. Glucose metabolism was altered likely as a consequence of obesity development. Finally, the authors also showed that central administration of CX3CL1 limits diet-induced weight gain in male mice. The authors conclude that microglial CX3CR1 signaling determine gender differences in obesity susceptibility.

The manuscript is well written, uses a deductive thread and thus is very easy to follow. The studies are well planned and conducted, and the results seem robust and clearly presented. The hypothesis is attractive and novel, and the authors use appropriate models to address it. So generally speaking, this is a very interesting study that addresses a relevant question in the energy homeostasis field. Despite these strengths there are a number of concerns and suggestions that the authors may want to address:

1. The literature shows that female mice are more resistant to diet-induced obesity. This is well illustrated by refs 5 and 6, in which females develop a milder obese phenotype when compared to males. The authors use a quite severe diet-induced protocol (60%, 18 weeks). Nevertheless, female mice show no changes at all in terms of body weight gain. So this is surprising, as other studies (including 5 and 6) show that females are less susceptible to obesity but not totally protected. The authors should comment on this.
2. Tissue weight (fat mass and lean mass) should be normalized by body weight.
3. A more extensive microglial analysis should be conducted in Fig1. By only measuring I11b, Ikbkb, cx3cl1 and cx3cr1 the authors conclude that females do not show microglial activation under HFD feeding conditions. Ibal staining and expression of other relevant cytokines should be also measured.
4. The authors state that lean mass was unaffected by HFD administration, but in fact Fig1d shows a significant reduction in both males and females. This should be corrected.
5. The length of HFD administration is not clear. In page 3 the authors state that is 18 weeks, but in pages 4 and 7 they state 3 months (12 weeks). Please, clarify these discrepancies.
6. In the conclusions, the authors state that they provide evidence of a neuron-microglia crosstalk. Although this is likely the case, the authors do not formally demonstrate it. Thus this conclusion should be softened.
7. In the methods section, the authors mention the MacGreen mouse which is actually not used in the present manuscript. This should be removed.
8. The authors focus on microglial activation in the hypothalamic region as the cause of obesity development. Have the authors assessed microglial activation in other relevant brain regions

involved in food intake control under HFD? The CX3CL1 pharmacological delivery route used by the authors is the lateral ventricle, thus these effects could be actually mediated by extra-hypothalamic microglia. The authors should provide more robust evidence that hypothalamic microglia is directly involved in the beneficial effects of CX3CL1 (i.e., injecting in the 3V which is closer to the mediobasal hypothalamus).

9. In previous studies, the authors showed POMC neuron loss, together with gliosis, under HFD conditions. This raises the possibility that excessive microglial activation causes neuronal degeneration. It would be interesting to measure POMC neuron number and integrity in HFD fed control and KO females. Furthermore, does microglial activation alter neuropeptide expression?

10. Alternative methods to induce microglial activation (such as LPS administration) would cause obesity? This is a sort of curiosity question, but would help to understand causality in this process.

Reviewers' comments:

Reviewer #1 (Remarks to the Author):

"Gender differences in microglial CX3CR1 signaling determine obesity susceptibility" by Dorfman et al. emphasized the CX3CL1/CX3CR1 as key modulator in DIO associated hypothalamic microgliosis and conclude that higher CX3CL1 input is the main reason that female mice are resistant to DIO. The study was performed carefully and conclusions were drawn properly. This is a timely study on the role played by the non-neuronal cells in regulation of energy metabolism.

We thank the reviewer for highlighting the important findings of our paper and acknowledging the potential interest for the readership of *Nature Communications* and the metabolism field in general. In response to reviewer concerns, we have added a considerable amount of new data, including evidence indicating that hypothalamic-targeted CX3CL1 reduces weight gain in HFD-fed male mice. This finding strengthens the manuscript by providing a direct demonstration that hypothalamic microglial activation is an important determinant of obesity susceptibility. In addition, we add new evidence suggesting estrogen is not the primary mediator of sex-specific protection from DIO through the CX3CR1 signaling system. Finally, we have made a number of improvements and changes throughout the manuscript based on the valuable feedback from the reviewers, as detailed below.

My major concern about this study is this study is descriptive without clear mechanism. In fact, authors have proposed several potential mechanistic experiments, that can be performed without

obvious obstacles, for example "whether female DIO resistance stems in part from the interaction of CX3CR1 signaling with estrogen, a potent anti-obesity hormone, that also suppresses microglial activation and inflammatory signaling", according to my understanding, the CX3CL1/CX3CR1 interaction with estrogen can be easily checked by *ex vivo* experiments.

We agree with Reviewer 1 that an understanding of the mechanism underlying the sex differences in CX3CR1 regulation of metabolism is an important issue worthy of further investigation. To address the specific possibility that estrogen mediates sex differences in CX3CR1 signaling, we have undertaken a series of studies using a variety of approaches (Supplemental Fig 6). First, we observed that levels of *Cx3cl1* and *Cx3cr1* mRNA in the hypothalamus do not differ between ovariectomized (OVX) mice and sham-operated controls (Supplemental Fig 6a-d). Similarly, pharmacologic administration of estrogen does not alter *Cx3cl1* and *Cx3cr1* gene expression (Fig. 6e-f). These data suggest that estrogen is not a significant physiologic regulator of the CX3CL1-CX3CR1 pathway.

To address the possible connection between estrogen and CX3CR1 signaling more definitively, we took two *in vivo* approaches. First, we performed a study examining the effect of estradiol (E2) replacement following OVX in CX3CR1 KO and heterozygous controls using a within subjects design (due to limitations with animal numbers). After OVX, mice were placed on a HFD for 9 weeks followed by 4 weeks of systemic estrogen replacement by continuous subcutaneous E2 infusion. As expected, OVX animals became obese and hyperphagic, but to the same extent in both genotypes (Fig 6g-h), indicating that DIO in the context of OVX is unaffected by CX3CR1 deficiency. This finding raises the possibility that estrogen deficiency promotes weight gain through inactivating microglial CX3CR1. Alternatively, loss of either estrogen or CX3CR1 signaling predisposes to DIO by engaging common downstream mechanisms such as increased production of microglial inflammatory mediators. Consistent with the latter hypothesis, body weight and food intake of the two groups were reduced to an equivalent degree by E2 replacement (Fig 6i-j). Together, these data suggest that estrogen and CX3CR1 protect against DIO via independent mechanisms. To further test this hypothesis using a different paradigm, we asked whether the well-documented anorexic effect of estrogen in male mice is dependent on intact CX3CR1 signaling. As predicted, a single injection of the long-acting estradiol benzoate (10mg sc) decreased 72h food intake but this effect did not differ between genotypes (Fig 6k). These data indicate that estrogen-mediated suppression of HFD feeding in male mice does not require CX3CR1 signaling, consistent with what was observed in female mice receiving E2 replacement following OVX.

These results collectively demonstrate that intact CX3CR1 signaling is required for protection against DIO in female mice but not male mice (Fig. 2) in an estrogen-independent manner. Although future studies are needed to identify the underlying mechanism in females, we report that HFD feeding reduces hypothalamic content of both ligand CX3CL1 (Fig. 1i) and receptor CX3CR1 (Fig. 1j) in wild-type male mice, rendering them functionally CX3CR1-deficient and therefore DIO susceptible. Thus, sensitivity of hypothalamic CX3CR1 signaling to HFD-induced disruption may underlie sex differences in DIO susceptibility, but this difference is not estrogen-mediated. Consistent with this interpretation, weight gain is reduced in males by central administration or hypothalamic overexpression of CX3CL1 (Fig. 4). We have revised the Discussion section of the manuscript to highlight these new findings and their implications for

future investigations of underlying mechanisms, including epigenetic, neural, chromosomal or immune-mediated sex differences in DIO susceptibility.

In addition, I have some minor issues:

1. Since most of the experiments were aimed to compare gender or genotype susceptibilities to diet (induced obesity) (interaction), two-way ANOVA but t-test should be performed for all data analysis involving two factors, especially for those in figure.1-3.

We agree with the reviewer that comparisons involving two factors or more require analysis by two-way ANOVA, which we had performed throughout the study including all graphs in fig 1-3. For example, there is a significant interaction between diet and gender in Fig. 1c and diet and genotype in 2f. We have elected not to report these F values, as they do not provide additional insight beyond the evident increase in fat mass in males (Fig. 1c) and CX3CR1 KO females (Fig. 2f) relative to controls. In contrast the genotype x time interaction in Fig. 4a indicates the overall effect of CX3CL1 infusion, which is more relevant than the individual time points. The asterisks throughout these figures and the rest of the manuscript where multiple factors are considered represent statistically significant post-hoc testing using Bonferroni corrections, and this is now made clear in the legend. The only data specifically excluded from this approach were the microglial *I11b*, *Ikbkb*, and *Cx3cr1* mRNA levels (Figs. 1g, 1h, and 1j) because the male and female tissues were extracted from different cohorts of mice processed on separate occasions for quantitative PCR, making them not directly comparable. We have now separated the two graphs of *Cx3cr1* data into Figs. 1j and 1k to avoid confusion.

2. In Fig.1i and j, basal CX3CL1 levels in Chow diet should not be normalized to 1 in female chow, this failed to show the difference of basal level between male and females. Whether females already had higher CX3CL1 secretion tone under physiological conditions which can prevent the DIO development should be presented. Also, same for the CX3CR1 levels. CX3CR1 protein levels have not been measured in either gender or diet. qPCR data failed to show the CX3CR1 basal mRNA level comparisons between male and female mice. Whether higher tone of CX3CR1 signaling and less *iba1* reactivity (in Fig.3 a-j) in female mice are mainly due to higher ligand input, or higher receptor expression, or both should be elaborated.

We thank the reviewer for this suggestion and have now replaced the original CX3CL1 data with absolute values expressed as ng/mg of tissue to allow a direct comparison between the sexes. While males have higher basal hypothalamic CX3CL1 levels than females, they respond to HFD with a decrease in ligand levels. Similarly, *Cx3cr1* mRNA decreases in males fed HFD, suggesting that coordinated decreases of both ligand and receptor reduce net CX3CR1 signaling and thereby undermine protection from DIO in males. By contrast, CX3CL1 and CX3CR1 levels are only modestly affected by HFD feeding in females, leaving them able to resist HFD-induced gliosis and weight gain.

3. In Fig.2 a and b, the absolute BW from wk0 should be presented, at least in suppl data.

We have provided the absolute BW data in Supplemental Fig 2 as requested. These curves look virtually identical to the delta curves in Fig. 2a, b due to the close matching of BWs at the start of the study.

4. The animal model used in this study is a global KO model. Though Fig.4 further tested brain specific effect by ICV infusion, ICV infusion from lateral ventricle could cause whole brain effect. In Fig.1, 3 and 4, at the molecular level, author only showed changes of iba1 immunoreactivity, CX3CL1/CX3CR1 and other gene mRNA levels in medial basal hypothalamus. Whether this global or whole brain phenotype had an exclusive link with medial basal hypothalamus is still questionable. This needs more supporting evidences or references included in the discussion. Besides, whether CX3CL1/CX3CR1 level differences between males and females only take place in MBH is another hole to be filled in Fig.1, or in suppl data. High magnification of iba1-ir images should be presented, to show whether there are differences on iba1-ir morphology, for example the number of processes and coverage of individual microglia.

We thank Reviewer 1 for these comments. First, the ICV infusion experiments show that in male mice, increasing whole-brain CX3CL1 action reduces both hypothalamic microgliosis and weight gain during HFD feeding (Fig. 4). These findings are consistent with published evidence linking hypothalamic microglia to the response to HFD exposure, including accumulation in the MBH along with reactive astrocytes (Thaler et al., *JCI* 2012; Buckman et al., *J Comp Neurol* 2013; Gao et al., *Glia* 2014; Valdearcos et al., *Cell Reports* 2014). In addition to providing information about effects of HFD and CX3CL1 action on hypothalamic microglial number and associated changes of inflammatory gene expression, we have added a new analysis of cell morphology focusing on microglial process length, which inversely correlates with activation state. Indeed, we observed reduced process length in female CX3CR1 KO mice fed HFD (but not female control mice) (Fig. 3k-m), consistent with the pattern of microglial accumulation and activation (Figs 3f-j and 3n). In addition to the new morphometric analysis, we performed additional studies to bolster our conclusions regarding the central action of CX3CL1-CX3CR1 signaling in metabolic regulation. Using stereotaxic intra-hypothalamic microinjection, we show that viral overexpression of CX3CL1 (Fig. 4i-l) confers to male mice a degree of protection against HFD-induced weight gain remarkably similar to that induced by ICV infusion of CX3CL1 protein (compare Figs. 4a and 4k).

Together, these data support the conclusion that CX3CR1 deletion in female mice undermines protection against DIO by enabling microglial responses that are normally limited by CX3CR1 signaling. Similarly, HFD-induced reductions of CX3CL1 and CX3CR1 in males predispose them to microgliosis and DIO, a phenotype that is partially reversible with increased CX3CL1 action in the hypothalamus. We submit that these findings represent an important advance in our understanding of obesity pathogenesis that will be of compelling interest to readers of *Nature Communications* even in the absence of a defined mechanism to explain how HFD feeding reduces hypothalamic CX3CR1 signaling in male, but not female mice. Our finding that the difference is not estrogen-mediated offers direction for future investigation of this key question.

5. Do neurons/hypothalamic neurons or astrocytes or any other cell populations from male and females have different CX3CL1 secretion levels under Chow and HFD? Does estrogen receptors expressed on the microglia?

With respect to the first question, CX3CL1 is highly expressed in the central nervous system (CNS), primarily by neurons (Cardona and Ransohoff, *Nature* 2010). Therefore, we measured levels of the secreted form of CX3CL1 in the whole hypothalamus in males and females fed

chow or HFD (Fig 1i), and report dynamic and sex-specific regulation of hypothalamic CX3CL1 levels by diet. While it is likely that this effect involves CX3CL1 derived from neurons, the lack of a high quality antibody precludes the cell-specific analysis needed to definitively identify the cellular source.

Regarding the second question, estrogen receptors are expressed on microglia (Sierra et al., *Glia* 2008; Saijo et al., *Cell* 2011), but whether CX3CL1 synthesis or secretion is regulated by ER signaling is unknown. Given the negative data reported above regarding a potential role for estrogen in the control of CX3CR1 signaling, we do not perceive a compelling need to answer this question in the current manuscript.

6. Whether restore CX3CR1 expression in microglia from HFD fed male brain could rescue the microgliosis?

While we enthusiastically agree with Reviewer 1 about performing microglial rescue experiments, the methodology necessary to genetically manipulate microglia *in vivo* unfortunately does not exist at this time. We and other groups are working to develop new tools for this purpose but have yet to achieve a sufficiently robust method to perform the relevant experiments. In particular, viral methods using adenovirus, AAV, and lentivirus all show efficacy *in vitro*, but inadequate infectivity *in vivo* to modify a sufficiently large population of cells. Similarly, adoptive transfer of microglial cells obtained from wild-type donors does not result in stable engraftment to allow for reliable subsequent analysis. Thus, we have elected to address the specific role of hypothalamic microglia using AAV-mediated overexpression of CX3CL1 in the MBH (AAVs infect neurons rather than microglia) (Fig. 4i-l). These new data indicate that CX3CL1 action in the MBH (likely resulting from increased neuronal synthesis of CX3CL1) is sufficient to reduce HFD-induced weight gain in male mice, consistent with a role for hypothalamic microglial activation in DIO susceptibility.

6. Fig.4 a-e showed preventative effect of central ICV infusion of CX3CL1 to DIO. Here vehicle groups are missing, at least a "HET+VEH" group should be included to show the CX3CL1 effect on WT.

While we agree with the reviewer that a VEH control is a necessary component of the central CX3CL1 experiments, we feel that the 4 separate protocols that included either a VEH-treated group with the DIO regimen (Fig. 4f-h) or a GFP group for the CX3CL1 hypothalamic AAV experiment (Fig. 4i-l) provide adequate controls for the overall effect of central CX3CL1 administration. Due to animal number limitations, we did not include VEH controls in the original prevention studies (4a-e and Supplemental fig 7), but instead opted to assess for off-target effects of CX3CL1 treatment using KO groups in both studies. Taken together, the 4 studies offer compelling evidence that hypothalamic CX3CL1 action in male mice is sufficient to reduce microgliosis and DIO susceptibility, presumably by compensating for the HFD-induced reduction in CX3CL1-CX3CR1 signaling.

7. All the glucose tolerance data included in the article was explained simply by fat mass difference. These data haven't been interpreted carefully. Whether glucose intolerance was due to fat mass difference needs to be tested in body weight matched studies, brain dominated effects

on glucose uptake could not be excluded either.

While we cannot definitively exclude a weight-independent effect of CX3CR1 deletion on glucose homeostasis, our finding that male CX3CR1 KO males fed HFD have no worsening of glucose homeostasis relative to HET controls (Fig. 2l and 2n) argues against this possibility. In response to the Reviewer's suggestion, however, we have adjusted the text of the Results section to acknowledge the need for additional studies to investigate the possibility of a direct contribution of CNS CX3CR1 signaling to glucose homeostasis.

Reviewer #2 (Remarks to the Author):

This manuscript focuses on the observation that female mice seem protected from the obesity inducing effects of a high fat diet. The hypothesis that activation of microglia in male hypothalamus but not female is the basis for the sex difference in response to a high fat diet is tested and largely confirmed. The notion that brain inflammation is a by product of obesity has been firmly established but there has been relatively little attention to sex differences in the impact of diet and how neuroinflammation might contribute to that difference. The use of a transgenic mouse lacking the fractalkine receptor and administration of the ligand to males creates a well balanced design that confirms differential roles for microglia in diet induced obesity in males and females. Given the human sex differences in obesity this work makes an important contribution to the literature.

There are some conclusions in the manuscript that are not well supported and some concerns about presentation.

Major Concerns.

1) Throughout the manuscript and even in the title the authors use the term gender when referring to the mice. Gender is an exclusively human construct that consists of both self and societies perception of ones sex. Animals do not have gender, only sex. Please replace the term throughout.

We thank the reviewer for clarifying the distinction between “gender” and “sex” and have adjusted the text accordingly throughout the manuscript. It is remarkable how often these terms are mistakenly interchanged in the literature, and we are happy not to contribute further to this confusion.

2) The weakest data in the manuscript are on the accumulation of microglia in the hypothalamus of high fat diet fed mice. The use of Iba1 fluorescence staining for counting is not ideal as the sensitivity is low, the signal fades with time and there is no counter stain by which precise anatomical locations can be determined. This would be much better done with DAB immunohistochemistry and proper stereology.

We agree with the reviewer that there are potential limitations to the use of Iba1 fluorescence for microglial quantification but respectfully submit that the conclusions of our study remain valid for the following reasons. First, we standardized our staining methods for all samples to assure

comparability, performed quantifications within 1-2 days of immunostaining, and used a DAPI counterstain to delineate borders (particularly between VMH and ARC) and to ensure that only nucleated cells were counted. In addition, we have previously performed validation studies comparing Iba1 to other microglial markers including CX3CR1-GFP, P2Y12, and Tmem119, and these data confirm that no significant differences exist between these approaches (data not shown). Likewise, we and others have previously published cell counts using Iba1 immunofluorescence in the context of HFD (Morseli et al. *Cell Reports* 2014; Valdearcos et al., *Cell Reports* 2014; Gao et al., *Glia* 2014; Lee et al., *AJP Endo and Metabolism* 2013), and many groups have employed this method in other domains of microglial investigation including those published in *Nature Communications* (for example, Szalay et al, 2016; Miyamoto et al, 2016; Peng et al, 2016; Masuda et al, 2014). For these reasons, we believe that our approach conforms to accepted standards for the field and that our results are therefore interpretable.

Second, since all microglia in HET and KO animals express GFP, we have reassessed microglial number in female mice using GFP fluorescence in response to the Reviewer's concern (Supplemental Fig. 4). Although there are slight differences in absolute numbers of cells identified with this method, the overall effects of diet and genotype on hypothalamic microglia are equivalent to those observed with Iba1 immunostaining (Compare Fig 3j to Supplemental Fig. 4e).

Finally, we include a new analysis of cellular process/dendrite length as an alternative assessment of microglial activation (Fig. 3 k-m). This analysis is based on the well-documented effect of activated microglial cells to become "amoeboid", characterized by retraction of processes and enlargement of the cell body. This approach independently confirms our finding that CX3CR1 KO females exhibit HFD-induced activation of microglia that is not evident in controls.

3) It would be even better if proliferation of microglia versus the vague term "accumulation" could be discerned by injecting animals with BrdU and colocalizing with Iba1. As is the term "microgliosis" cannot be used since this has not been actually shown.

While we agree that it would be ideal to be able to distinguish microglial proliferation from cellular recruitment, this is a technically challenging undertaking due to the low numbers of BrdU+ cells predicted to account for the additional microglia observed in HFD-fed males and CX3CR1 KO females. In response to the Reviewer's suggestion, however, we nevertheless performed a pilot analysis using the BrdU method. Our data do not reveal significant numbers of replicating cells in the MBH of HFD-fed males (data not shown), but we hesitate to draw strong conclusions given the low sensitivity of this method.

Apart from this technical limitation, we add that the term "microgliosis" does not imply microglial proliferation. Instead, it is synonymous with microglial accumulation in response to CNS insult regardless of cause (which includes local self-renewal, recruitment of circulating macrophages, and differentiation from latent progenitors; see Streit et al., *Prog Neurobiol* 1999; Cardona and Ransohoff, *Nature* 2010). While a more complete understanding of this cellular response continues to evolve (Ajami et al., *Nat Neurosci* 2007; Elmore et al., *Neuron* 2014;

Lawson et al, Neuroscience 1992), the general use of the term has remained, and we have followed this standard.

4) The diagram showing the outline of the MBH suggests the focus is on the arcuate, but the MBH includes the VMN as well. A much more precise anatomical characterization of the microglia would enhance the manuscript.

We thank the reviewer for this suggestion. In response, as described above, we note that we performed the immunohistochemical analyses using DAPI staining which enables us to delineate the VMH from arcuate nucleus. Based on these and other data (Thaler et al, *JCI* 2012), there is minimal evidence of microglial accumulation or activation in the VMN or other hypothalamic areas in response to HFD. However, for consistency with the qPCR data, which represents all MBH microglia, we have opted to retain the more inclusive term. To clarify this point, we have now included a sentence in the Methods section describing the distinction between the areas within the MBH and the indication that microglial changes are primarily observed in the arcuate.

5) What is the rationale for standardizing mRNA levels to control in Figure 4K? PCR is a highly quantitative technique and so no % change transformation should be required.

For quantitative PCR experiments, we normalized CT values to 18-S controls followed by establishment of the HT-chow condition as the basis (100%) for determining fold-change. We have elected to include this HT-chow bar in the graph for each gene so information about variance within the control group is readily apparent. This method of data presentation avoids inappropriate comparisons between samples obtained from different cohorts at different times as well as highlights the relevant directionality and degree of change without obscuring the data with widely variable absolute values (which do not scale well in a single graph). We note that this approach to normalizing data from the experimental to the control group is used extensively in the literature, including recent papers published in *Nature Communications* (e.g. Whittle et al, 2015; Suberbielle et al, 2015; Minett et al., 2015; Cimino et al., 2016; J Lee et al., 2016).

6) Doesn't the data in Figure 4 suggest that the effect of the fractalkine receptor activation is to modulate food intake? This seems contrary to the notion that it is metabolism and glucose regulation that is the primary driver of weight gain. It would seem the central hypothesis that microglia activation regulates high fat diet induced obesity would have to be modified to microglia activation regulates appetite for high fat food. There is no effect on food intake in Figure 2 using the KO mouse, so this aspect of the manuscript is very confusing, are the microglia regulating metabolism or food intake?

We acknowledge that this is a complex issue, in part because HFD-induced weight gain is itself due to both increased caloric intake and decreased metabolic rate, and reduced CX3CR1 signaling is superimposed on this effect. We observed on the one hand that the obese phenotype of CX3CR1 KO females on a HFD was associated with reduced energy expenditure and no change in food intake relative to HET controls, while on the other hand central CX3CL1 administration (via ICV infusion or viral overexpression) to males reduced food intake (the impact on energy expenditure was not assessed). Thus, it is conceivable that central CX3CL1 treatment affects both metabolic rate and food intake; which of these predominates may depend

on the experimental context. Beyond this, we note that the failure of female CX3CR1 KO mice to reduce food intake in the face of lower energy expenditure and excess weight can be seen as contributing to the resultant obesity, potentially linking the two mechanisms. Alternatively, the connection between CX3CL1-CX3CR1 signaling and control of energy intake and expenditure may differ between sexes, but in either case it is clear that this signaling system protects against DIO, and that disruption of this system during HFD feeding predisposes male mice to obesity. Future studies are needed to delineate the precise impact of microglial activation on energy balance, its relationship to diet composition, and the extent to which the effect differs between males and females. In response to the Reviewer's concern, we have enhanced the Discussion section to highlight the discrepant nature of effects on energy intake and expenditure.

7) The data in Figure 4i are taken to show fat males can lose weight if injected with fractalkine ligand but the line with the asterisks is over the animals that received vehicle and gained weight relative to the start. It is not clear what the statistical comparison is here and if it is between or within groups.

We thank the Reviewer for pointing out the lack of clarity in this figure. As is the case throughout the manuscript when more than 2 factors are present (e.g. genotype and time in 4i), the asterisks reflect statistical significance ($p < 0.05$) between vertically-aligned points on the curve using post-hoc Bonferroni testing. The placement of the asterisks above the upper line is for visual clarity only (similar for Figs. 4a and 4k). Specifically, the data show a between-groups comparison demonstrating that the body weight change from the obese baseline significantly differs between CX3CL1-treated and vehicle-treated mice after about 2 weeks of treatment. We have added text clarifying this point in the Results section.

8) What is missing from the manuscript is any indication of the origin of the observed sex difference. The authors state there is no difference in female gonadal function in the KO mice, but what happens if the gonads are removed in both males and females to remove gonadal steroids? Would the sex difference go away? This simple step would provide major insight as to whether the observed effects are the result of differences in adult hormonal milieu or are the result of either developmental effects of hormones or chromosome complement.

We agree that this is a fundamental question, and it was raised by Reviewer 1 as well. In response to this concern, we performed an extensive series of studies to test the hypothesis that estrogen is responsible for the sex differences associated with CX3CL1-CX3CR1 signaling (Supplemental Fig. 6). As detailed in our response to the first comment raised by Reviewer 1, these studies provide clear evidence that neither the obesogenic effect of estrogen deficiency (induced by OVX) nor the effect of estrogen to cause weight loss are influenced by reduced CX3CR1 signaling. Thus, we conclude that differences in estrogen levels cannot explain sex differences in CX3CL1-CX3CR1 signaling, and future studies are needed to identify the underlying mechanism, be it developmental, epigenetic, immunologic, or chromosomal in nature. We hope the Reviewer will agree that insight into links between CX3CL1-CX3CR1 signaling and obesity pathogenesis constitute an important advance even if differences in estrogen levels do not explain how HFD feeding reduces hypothalamic CX3CR1 signaling in male, but not female mice.

Minor Concerns

1) In the Introduction the authors refer to microglia being maintained in a quiescent state at baseline but this term is no longer used, instead these microglia are surveying as they are actively interacting with nearby neurons all the time and appear to never rest.

We agree that our evolving understanding of microglial function indicates that microglia are not inactive or “quiescent” at baseline, but instead maintain active surveillance of the CNS parenchyma, and we thank the Reviewer for pointing this out. Yet it is also clear that constitutively produced neuronal factors such as CX3CL1 and CD200 constrain microglial activity in the basal state, thereby limiting production of neurotoxic mediators and phagocytosis. In other words, even though microglia are not quiescent *per se*, neurons play a key role to prevent their activation under basal conditions. To clarify this point, we altered the sentence of the Introduction containing the word “quiescent.” It now reads: “Under basal conditions, microglial activation is inhibited by several broadly-expressed neuronal factors including the chemokine CX3CL1 (also referred to as fractalkine), a cleavable transmembrane protein that binds the G_i-protein coupled receptor CX3CR1.”

2) It is surprising that the act of isolating microglia from the brain did not in and of itself activate them. Others have suggested that this approach cannot be used for assessing short-term signaling in microglia. Can the authors provide any insight as to why they have been successful where others have apparently failed?

While we agree with the Reviewer that microglial isolation and *ex vivo* analysis is technically challenging, the Percoll-based methodology that we employed has been used successfully and widely in microglia literature (Cardona et al, *Nat Neurosci* 2006; Chiu et al., *Cell Reports* 2013; Grabert et al., *Nat Neurosci* 2016; Butovsky et al, *Nat Neurosci* 2014; Hickman et al, *Nat Neurosci* 2013). Though two of the most recent 4 of these studies incorporated magnetic bead immunoaffinity sorting along with Percoll (Chiu et al., and Grabert et al.) while the other two used Percoll alone (Butovsky et al., and Hickman et al.), all 4 studies reported remarkably similar transcriptional profiles for “resting” microglia that were distinct from the gene expression of activated cells (due to aging, LPS or ALS). Thus, despite the possibility of *ex vivo* microglial activation, these techniques can reliably identify state-specific changes in microglial gene expression.

3) Pg 4 - last sentence, the phrase "overall lower microglial silencing in males" is both awkward and inaccurate, there is no evidence microglia are silent in females, they are just not reactive. The next paragraph again uses the term "microglia silencing", this terminology should be removed throughout.

We appreciate the clarification and have revised the manuscript to eliminate use of the term “silencing”. In the first case, we have substituted the highlighted phrase above with “reduced CX3CL1-CX3CR1 signaling in males.” In the subsequent paragraph, the text now reads “we hypothesized that intact CX3CR1 signaling in females during HFD feeding (Fig. 1i) limits

microglial reactivity and reduces obesity susceptibility.”

Reviewer #3 (Remarks to the Author):

In this study, Dorfman and colleagues investigated the gender susceptibility to obesity development in the context of microglial CX3CR1 signaling. They show that, in contrast to males, female C57Bl/6 mice are resistant to develop diet-induced obesity. This is associated with reduced hypothalamic microglial activation and increased content of CX3CL1. These results encouraged the authors to assess in more detail the gender effects of CX3CR1 signaling using a model deficient (KO) for this receptor. Interestingly, female KO mice fed with high-fat diet developed obesity to a similar extent than male mice. This "male-like" phenotype observed in KO females, was associated with increased adiposity in the face of normal food intake and reduced energy expenditure. Glucose metabolism was altered likely as a consequence of obesity development. Finally, the authors also showed that central administration of CX3CL1 limits diet-induced weight gain in male mice. The authors conclude that microglial CX3CR1 signaling determine gender differences in obesity susceptibility.

The manuscript is well written, uses a deductive thread and thus is very easy to follow. The studies are well planned and conducted, and the results seem robust and clearly presented. The hypothesis is attractive and novel, and the authors use appropriate models to address it. So generally speaking, this is a very interesting study that addresses a relevant question in the energy homeostasis field. Despite these strengths there are number of concerns and suggestions that the authors may want to address:

1. The literature shows that female mice are more resistant to diet-induced obesity. This is well illustrated by refs 5 and 6, in which females develop a milder obese phenotype when compared to males. The authors use a quite severe diet-induced protocol (60%, 18 weeks). Nevertheless, female mice show no changes at all in terms of body weight gain. So this is surprising, as other studies (including 5 and 6) show that females are less susceptible to obesity but not totally protected. The authors should comment on this.

We agree with Reviewer 3 that the female C57BL6 WT and CX3CR1 HET mice we obtained from Jackson Laboratories and bred in our facility are apparently less susceptible to HFD-induced weight gain than some—but not all (e.g. Perez-Sieira et al., *PLOS ONE* 2013)—female mice reported in the literature. Although our studies were not designed to investigate this difference, there are several reasonable possibilities. First, the mice were single housed throughout the HFD period, an approach that is often avoided due to cost considerations. Although single housing enables precise measurements of food intake and metabolic rate, it diminishes weight gain relative to group housing owing to the greater energy costs related to thermogenesis of animals housed alone. Second, small differences in strain purity or microbiome (based on our facility) can result in significant variation in DIO susceptibility (Abu-Toamih Atamni et al., *BMC Genet* 2016; Ussar et al., *Cell Metabolism* 2015). Third, we randomly assigned littermate mice to diet conditions using no specific inclusion or exclusion criteria, which may differ from studies where diet-sensitive animals or non-age matched controls may have been selected to facilitate the assessment of DIO resistance phenotypes. Finally, we used littermate controls throughout, ensuring that the internal comparisons of diet x genotype x sex

are valid. In response to the Reviewer's question, we have added a sentence in the Results summarizing these points.

2. Tissue weight (fat mass and lean mass) should be normalized by body weight.

We thank the Reviewer for this point and agree that both absolute and normalized values are important to report. We have now included the fat mass and lean mass percentages in Supplemental Fig. 3.

3. A more extensive microglial analysis should be conducted in Fig1. By only measuring Il1b, Ikbkb, cx3cl1 and cx3cr1 the authors conclude that females do not show microglial activation under HFD feeding conditions. Ibal staining and expression of other relevant cytokines should be also measured.

We agree with the Reviewer and point out that in Fig. 1, we included Il1b and Ikbkb because these inflammatory markers increased in the males, allowing a comparison of microglial reactivity between the sexes. We also include both the Ibal analysis and more extensive microglial profile for females in Fig. 3 (note that HET and WT females show no differences in these parameters (data not shown)). We feel this arrangement of the data highlights the effect of varying genotype and diet on sex-specific differences in microglial accumulation and activation, but can change the format if recommended by the Reviewer.

4. The authors state that lean mass was unaffected by HFD administration, but in fact Fig1d shows a significant reduction in both males and females. This should be corrected.

We thank the Reviewer for catching this inadvertent error. We have corrected the text to indicate the small but significant reduction in lean mass with HFD administration.

5. The length of HFD administration is not clear. In page 3 the authors state that is 18 weeks, but in pages 4 and 7 they state 3 months (12 weeks). Please, clarify these discrepancies.

We appreciate the Reviewer's pointing out this confusing information. The difference in diet length reflects the fact that these three experiments were performed on distinct cohorts of mice, and the duration of HFD feeding was not fixed across each of the studies. We have clarified this point in the text.

6. In the conclusions, the authors state that they provide evidence of a neuron-microglia crosstalk. Although this is likely the case, the authors do not formally demonstrate it. Thus this conclusion should be softened.

We agree that our data do not directly address neuron-microglia interactions. Therefore, we have removed the references to "neuron-microglia crosstalk" and instead focused exclusively on microglial activation and CX3CR1 signaling.

7. In the methods section, the authors mention the MacGreen mouse which is actually not used in the present manuscript. This should be removed.

The methods section has been corrected with the MacGreen mouse removed.

8. The authors focus on microglial activation in the hypothalamic region as the cause of obesity development. Have the authors assessed microglial activation in other relevant brain regions involved in food intake control under HFD? The CX3CL1 pharmacological delivery route used by the authors is the lateral ventricle, thus these effects could be actually mediated by extra-hypothalamic microglia. The authors should provide more robust evidence that hypothalamic microglia is directly involved in the beneficial effects of CX3CL1 (i.e., injecting in the 3V which is closer to the mediobasal hypothalamus).

We agree that this is a key point, which was also raised by Reviewer 1. As described in more detail in our response to Reviewer 1, we have added new data showing that viral overexpression of CX3CL1 in the mediobasal hypothalamus (MBH) causes weight reduction in HFD-fed male mice. Together with the highly similar degree of DIO protection induced by lateral ventricle infusion (Fig 4k vs 4a), our data strongly suggest that action of CX3CL1 in the MBH is sufficient to suppress local microglial activation and thereby protect against DIO. This interpretation is consistent with the observation that HFD-induced alterations to microglia appear limited to the MBH (Thaler et al., *JCI* 2012; Gao et al., *Glia* 2014; Valdearcos et al., *Cell Reports* 2014). Nevertheless, differences in food intake reduction between ICV CX3CL1 treatment and viral hypothalamic CX3CL1 overexpression raise the possibility of additional brain sites of CX3CR1-mediated regulation of energy homeostasis, a subject of future investigation.

9. In previous studies, the authors showed POMC neuron loss, together with gliosis, under HFD conditions. This raises the possibility that excessive microglial activation causes neuronal degeneration. It would be interesting to measure POMC neuron number and integrity in HFD fed control and KO females. Furthermore, does microglial activation alter neuropeptide expression?

This is a very interesting point and we agree that an assessment of POMC neurons in this context is of interest, although we note that a more recent study from our group (Berkseth et al., *Endocrinology* 2014) did not find changes in POMC neuron number at 5 months of HFD feeding (rather than the 8 months in our original *JCI* paper). Given the 3-4 month time frame of the studies we conducted, we are concerned that a longer duration of HFD feeding is needed to reliably detect POMC neuron injury/loss and hence have elected to defer this analysis to a subsequent long-term study. Also relevant to this decision is the fact that the tissue processing method for microglial isolation and immunohistochemistry used in the current study precludes POMC cell-specific analyses and measurement of neuropeptide expression.

10. Alternative methods to induce microglial activation (such as LPS administration) would cause obesity? This is a sort of curiosity question, but would help to understand causality in this process.

The impact of different stimuli that elicit microglial activation on energy homeostasis is an important and complex question that we have discussed in a previous review (Thaler et al., *Front Neuroendocrinol* 2010). The primary limitation of these comparisons is that most microglial

activators cause pleiotropic effects in many cell types. For example, LPS induces inflammatory signaling in endothelial cells, astrocytes, neurons, and other cell types, confounding interpretation of its effects on microglia and metabolism. Furthermore, sickness-induced microglial activation (e.g., by LPS) promotes a much more rapid, widespread, and severe CNS inflammatory response than the low-grade, chronic hypothalamic inflammation elicited by HFD feeding in male mice. Thus, resolution of this paradox requires cell-specific methodologies that have only recently been developed and are a subject of current investigation in our lab and elsewhere.

Reviewer #1 (Remarks to the Author)

In the revision and rebuttal, all the points I have raised have been well addressed. I have no more comments. The manuscript is suitable for publishing in Nature Communication. And I wish future studies by authors or other researchers would answer the questions on how estrogen and Cx3cr1 protect against DIO.

Reviewer #2 (Remarks to the Author)

the authors have done an excellent job of responding to the many diverse comments of the reviewers and provide additional data in multiple instances. I was pleased to see the dose of estradiol used in males was 10 micrograms as stated in the manuscript, the rebuttal letter stated 10 milligrams which would likely be toxic. A minor point is the use of estrogen versus estradiol. The actual hormone is estradiol and it is a type of estrogen. The term estrogen is used in place of estradiol in several instances and so the authors might want to correct this

Reviewer #3 (Remarks to the Author)

The authors have addressed my concerns. I do not have further questions.

REVIEWERS' COMMENTS:

Reviewer #1 (Remarks to the Author):

In the revision and rebuttal, all the points I have raised have been well addressed. I have no more comments. The manuscript is suitable for publishing in Nature Communication. And I wish future studies by authors or other researchers would answer the questions on how estrogen and Cx3cr1 protect against DIO.

We thank the reviewer for their help throughout the process.

Reviewer #2 (Remarks to the Author):

the authors have done an excellent job of responding to the many diverse comments of the reviewers and provide additional data in multiple instances. I was pleased to see the dose of estradiol used in males was 10 micrograms as stated in the manuscript, the rebuttal letter stated 10 milligrams which would likely be toxic. A minor point is the use of estrogen versus estradiol. The actual hormone is estradiol and it is a type of estrogen. The term estrogen is used in place of estradiol in several instances and so the authors might want to correct this

We thank the reviewer for their help throughout the process. We have adjusted the text to read "estradiol" where infusions or injections were performed to clarify the distinction from the general action of estrogen or the effect of estrogen deficiency (e.g. OVX).

Reviewer #3 (Remarks to the Author):

The authors have addressed my concerns. I do not have further questions.

We thank the reviewer for their help throughout the process.